# Global tropospheric halogen (Cl, Br, I) chemistry and its impact on oxidants

Xuan Wang[1,2], Daniel J. Jacob[3], William Downs[3], Shuting Zhai[4], Lei Zhu[5], Viral Shah[3], Christopher D. Holmes[6], Tomás Sherwen[7,8], Becky Alexander[4], Mathew J. Evans[7,8], Sebastian D. Eastham[9], J. Andrew Neuman[10,11], Patrick R. Veres[10], Theodore K Koenig[11,12], Rainer Volkamer[11,12], L. Gregory Huey[13], Thomas J. Bannan[14], Carl J. Percival[14,a], Ben H. Lee[4], and Joel A. Thornton[4]

[1] School of Energy and Environment, City University of Hong Kong, Hong Kong SAR, China
[2] City University of Hong Kong Shenzhen Research Institute, Shenzhen, China
[3] School of Engineering and Applied Sciences, Harvard University, Cambridge, Massachusetts, USA
[4] Department of Atmospheric Sciences, University of Washington, Seattle, USA
[5] School of Environmental Science and Engineering, Southern University of Science and Technology, Shenzhen, China
[6] Department of Earth, Ocean, and Atmospheric Science, Florida State University, Tallahassee, Florida, USA
[7] Wolfson Atmospheric Chemistry Laboratories, Department of Chemistry, University of York, York, UK
[8] National Centre for Atmospheric Science, University of York, York, UK
[9] Laboratory for Aviation and the Environment, Massachusetts Institute of Technology, Cambridge, Massachusetts, USA
[10] NOAA Chemical Sciences Laboratory (CSL), Boulder, Colorado, USA
[11] Cooperative Institute for Research in Environmental Sciences, University of Colorado, Boulder, Colorado, USA
[12] Department of Chemistry, University of Colorado, Boulder, CO, USA
[13] School of Earth and Atmospheric Science, Georgia Institute of Technology, Atlanta, Georgia, USA
[14] School of Earth, Atmospheric and Environmental Sciences, University of Manchester, Manchester, UK
[a] Now at: Jet Propulsion Laboratory, California Institute of Technology, Pasadena, California, USA

*Correspondence to*: Xuan Wang (xuanwang@cityu.edu.hk)

**Abstract.** We present an updated mechanism for tropospheric halogen (Cl + Br + I) chemistry in the GEOS-Chem global atmospheric chemical transport model and apply it to investigate halogen radical cycling and implications for tropospheric oxidants. Improved representation of HOBr heterogeneous chemistry and its pH dependence in our simulation leads to less efficient recycling and mobilization of bromine radicals, and enables the model to include mechanistic sea salt aerosol debromination without generating excessive BrO. The resulting global mean tropospheric BrO mixing ratio is 0.19 ppt, lower than previous versions of GEOS-Chem. Model BrO shows variable consistency and biases in comparison to surface and aircraft observations in marine air, which are often near or below the detection limit. The model underestimates the daytime measurements of $Cl_2$ and BrCl from the ATom aircraft campaign over the Pacific and Atlantic, which if correct would imply a very large missing primary source of chlorine radicals. Model IO is highest in the marine boundary layer and uniform in the free troposphere, with a global mean tropospheric mixing ratio of 0.08 ppt, and shows consistency with surface and aircraft observations. The modeled global mean tropospheric concentration of Cl atoms is 630 cm$^{-3}$, contributing 0.8% of the global oxidation of methane, 14% of ethane, 8% of propane, and 7% of higher alkanes. Halogen chemistry decreases the global tropospheric burden of ozone by 11%, $NO_x$ by 6%, and OH by 4%. Most of the ozone decrease is driven by iodine-catalyzed loss. The resulting GEOS-Chem ozone simulation is unbiased in the Southern Hemisphere but too low in the Northern Hemisphere.

## 1 Introduction

Halogen radicals (chlorine, bromine, iodine) have a broad range of implications for tropospheric oxidant chemistry. They originate from sea salt aerosols (SSA), emitted halogen gases, and transport from the stratosphere, and cycle rapidly with inorganic non-radical reservoirs (Platt and Hönninger, 2003; Finlayson-Pitts, 2003; Saiz-Lopez and von Glasow, 2012; Simpson et al., 2015;

Wang et al., 2019). Cl, Br, and I atoms provide sinks for volatile organic compounds (VOCs), dimethylsulfide (DMS), and mercury (Atkinson, 1997; Saiz-Lopez and von Glasow, 2012; Horowitz et al., 2017). Cycling between halogen radicals and their reservoirs converts $NO_x$ to $HNO_3$ and causes catalytic loss of ozone (von Glasow et al., 2004; Yang et al., 2005; Sherwen et al., 2016b). Reaction of $Cl^-$ with $N_2O_5$ in polluted environments at night produces $ClNO_2$ that photolyzes in the daytime to return Cl atoms and

$NO_2$, stimulating ozone production (Osthoff et al., 2008; Roberts et al., 2008). Acid displacement of $Cl^-$ by $HNO_3$ is a source of $NO_3^-$ aerosol. Reviews by Saiz-Lopez and von Glasow (2012) and Simpson et al. (2015) describe this fundamental knowledge of tropospheric halogen chemistry in more detail.

A number of global modelling studies have explored the importance of halogen chemistry in the troposphere (von Glasow et al.,
2004; Saiz-Lopez et al., 2006; Ordóñez et al., 2012; Long et al., 2014), but there remain large uncertainties in sources and chemical mechanisms. Here we present a new mechanistic description of halogen tropospheric chemistry in the GEOS-Chem global model that synthesizes previous GEOS-Chem developments (Parrella et al., 2012; Eastham et al., 2014; Schmidt et al., 2016; Sherwen et al., 2016a;b; Sherwen et al., 2017; Chen et al., 2017; Wang et al., 2019; Zhu et al., 2019) and includes a number of updates. We use the updated model to interpret recent observations of tropospheric halogens, describe halogen radical cycling, and quantify the
impacts on tropospheric oxidant chemistry. Shah et al. (2021) examine the impact of our simulated Br and Cl atom concentrations in a new redox mechanism for atmospheric mercury.

## 2 Tropospheric halogen chemistry in GEOS-Chem

We describe here our updated representation of tropospheric halogen chemistry in version 12.9 of GEOS-Chem (http://www.geos-chem.org), implemented as part of the general model mechanism for coupled ozone–$NO_x$–VOCs–aerosol–halogen tropospheric
and stratospheric chemistry. Extensive referencing will be made to Sherwen et al. (2016b), who implemented the previous representation of tropospheric halogen chemistry in GEOS-Chem (version 11-02), and to Wang et al. (2019), who described an earlier version of the mechanism implemented here. GEOS-Chem stratospheric halogen chemistry is as described by Eastham et al. (2014) and we will not discuss it further here.

### 2.1 Sources of tropospheric halogens

Table 1 lists the global sources and sinks of tropospheric gas-phase inorganic chlorine ($Cl_y$), bromine ($Br_y$), and Iodine ($I_y$) in GEOS-Chem (see Table 1 for definitions of $Cl_y$, $Br_y$, and $I_y$). SSA emissions are from Jaeglé et al. (2011). Open fire emissions of HCl are obtained by applying the emission factors from Andreae (2019) for different vegetation types to the GFED4 (Global Fire Emissions Database version 4) biomass burned inventory (van der Werf et al., 2017). The resulting global source of 0.5 Tg Cl $a^{-1}$ is much smaller than in Wang et al. (2019), who used older emission factors from Lobert et al. (1999). Organohalogen gases can
produce halogen radicals by oxidation and photolysis. Emissions of $CH_3Cl$, $CH_2Cl_2$, $CHCl_3$, $CHBr_3$ are implicitly treated in the model by specifying latitudinally and monthly surface air boundary conditions from CMIP6 (Historical greenhouse gas concentrations for climate modelling) (Meinshausen et al., 2017). Emissions of other bromocarbons ($CH_3Br$, $CH_2Br_2$) and iodocarbons ($CH_3I$, $CH_2I_2$, $CH_2ICl$, $CH_2IBr$) are from Bell et al. (2002), Liang et al. (2010), and Ordóñez et al. (2012).

We do not include continental emissions of inorganic chlorine from anthropogenic sources (fuel combustion, waste incineration, etc.) and dust, because they are highly uncertain and most likely negligible from a global perspective. The only available global

emission inventory for anthropogenic HCl and Cl⁻ is that of McCulloch et al. (1999) at 6.7 Tg Cl a$^{-1}$ for 1990s, but we previously found this inventory to be too high by an order of magnitude in comparison to regional inventories and atmospheric observations (Wang et al., 2019). Analysis of deposition data by Haskins et al. (2020) find that anthropogenic chlorine emissions have decreased by 95% in US since 1998, further indicating that the McCulloch et al. (1999) inventory is outdated. Our previous model comparisons to aerosol Cl⁻ observations indicate that anthropogenic chlorine sources are important in China (Wang et al., 2020), but not in the US where the observed Cl⁻ concentrations can be attributed to long-range transport of SSA plus some dust influence in the Southwest (Wang et al., 2019). Zhai et al. (2021), who include anthropogenic HCl emissions using observed HCl:SO$_2$ ratio (Lee et al., 2018), also find anthropogenic sources of chlorine is very small over North America and western Europe. Because of this neglect of anthropogenic sources, our model results may underestimate chlorine concentrations in continental source regions.

The main global source of tropospheric Cl$_y$ is mobilization of Cl⁻ from SSA. A total of 50 Tg Cl⁻ a$^{-1}$ (2.4% of SSA emissions) is mobilized to Cl$_y$ in the model by acid displacement and other heterogeneous reactions. This number is smaller than our previous estimate in Wang et al. (2019) (64 Tg Cl⁻ a$^{-1}$), mainly due to slower ClNO$_2$ generation from the N$_2$O$_5$ + Cl⁻ reaction (Section 2.3). Organochlorines provide a tropospheric source of 3.3 Tg Cl⁻ a$^{-1}$ as Cl atoms from photolysis and oxidation. Transport from stratosphere adds 0.14 Tg Cl a$^{-1}$ to tropospheric Cl$_y$. The source of I$_y$ is estimated to be 2.7 Tg I a$^{-1}$, mostly from the inorganic iodine (HOI, I$_2$) formed from the ocean surface reaction of O$_3$ with iodide (I⁻), based on Carpenter et al. (2013) and MacDonald et al. (2014), and as described by Sherwen et al. (2016b).

In GEOS-Chem versions before 12.9, SSA debromination was not included despite being known to be an important source for Br$_y$ (Sander et al., 2003). This is because SSA debromination generated excessive BrO concentrations in comparison to observations, which then drove excessive ozone depletion (Schmidt et al., 2016;Zhu et al., 2019). Revision of HOBr reactive uptake as source of bromine radicals effectively corrects this problem (Section 2.2), allowing us to include mechanistically the SSA debromination source. This provides the main global source of tropospheric Br$_y$ (20 Tg Br a$^{-1}$), mostly through the HOBr/HOCl/HOI + Br⁻ heterogeneous reactions. Bromocarbon gases contribute only 0.54 Gg Br a$^{-1}$ to Br$_y$ but still dominate the Br$_y$ source in the free troposphere.

**2.2 Chemical mechanism**

Our tropospheric halogen chemistry mechanism synthesizes and updates previous GEOS-Chem mechanistic developments. Chlorine chemistry in GEOS-Chem was first built in Eastham et al. (2014) for the stratosphere and extended to troposphere by Schmidt et al. (2016), with updates by Sherwen et al. (2016b), Sherwen et al. (2017), Wang et al. (2019), and Wang et al. (2020). Tropospheric bromine chemistry was first built by Parrella et al. (2012) with updates to heterogeneous reactions by Schmidt et al. (2016), Chen et al. (2017), Wang et al. (2019), and Zhu et al. (2019). Iodine chemistry was built by Sherwen et al. (2016a) and Sherwen et al. (2016b). Recent general model updates important for halogen chemistry include a new method of simulating cloud chemistry in partly cloudy grid cells that accounts for limitation by entrainment of air into the cloud (Holmes et al., 2019) and an improved cloudwater pH calculation that considers carboxylic acids and dust alkalinity (Moch et al., 2020; Shah et al., 2020). Aqueous aerosol thermodynamics including calculation of aerosol pH and HCl/Cl⁻ partitioning are from ISORROPIA II (Fountoukis and Nenes, 2007).

We update here the reactive uptake of HOBr by aerosols and cloud droplets (Table 2). This uptake which involves reactions with Br⁻, Cl⁻, and dissolved $SO_2$ ($S(IV) \equiv HSO_3^- + SO_3^{2-}$):

$$HOBr(aq) + Br^- + H^+ \rightarrow Br_2 + H_2O \qquad (R1)$$
$$HOBr(aq) + Cl^- + H^+ \rightarrow BrCl + H_2O \qquad (R2)$$
$$HOBr(aq) + HSO_3^- \rightarrow HBr + HSO_4^- \qquad (R3)$$
$$HOBr(aq) + SO_3^{2-} \rightarrow HBr + SO_4^{2-} \qquad (R4)$$

Reactions (R1) and (R2) with subsequent fast photolysis of $Br_2$ and BrCl recycle bromine radicals from HOBr and further mobilize Br⁻ and Cl⁻ to produce new radicals. In GEOS-Chem, the rates are applied to the following stoichiometry:

$$HOBr(aq) + YBr^- + (1 - Y)\, Cl^- + H^+ \rightarrow YBr_2 + (1 - Y)BrCl + H_2O \qquad (R5)$$

where $Y$ is the yield of $Br_2$ and 1-$Y$ is the yield of BrCl, which are calculated based on the laboratory study of Fickert et al. (1999) and described in Table 2.

Total reactive uptake of HOBr from reactions (R3)-(R5) in aqueous aerosols and clouds is calculated with a standard first-order reactive uptake coefficient γ (Jacob, 2000), calculated following Ammann et al. (2013):

$$\gamma = \left(\frac{1}{\Gamma} + \frac{1}{\alpha_b}\right)^{-1} \qquad (1)$$
$$\Gamma = 4H_{HOBr}RTI_r k^I f(r, I_r)/c \qquad (2)$$
$$I_r = \sqrt{D_l/k^I} \qquad (3)$$
$$k^I = k_3^I + k_4^I + k_5^I \qquad (4)$$

where $H_{HOBr}$ is the the Henry's law constant of HOBr (Sander, 2015), $T$ is temperature, $R$ is the universal gas constant (8.314 J K⁻¹ mol⁻¹); $D_l$ is the liquid phase diffusion coefficient for HOBr ($1.4\times10^{-5}$ cm²s⁻¹); $f(r, I_r)$ is the reacto-diffusive correction term, and $k^I$ is the first-order total reaction rate constant of HOBr from pathways (R3-R5) computed as a function of the concentrations of Br⁻, Cl⁻, H⁺, $HSO_3^-$, and $SO_3^{2-}$. After computing the overall loss of HOBr, we distribute the loss by pathways on the basis of the relative reaction rates $k_i^I$. Reactions (R3) and (R4) are important only in clouds where high liquid water content and relatively high pH enable dissolution of $SO_2$

Wang et al. (2019) previously calculated $k_5^I$ based on experimental results over limited and inconsistent pH ranges (pH = 1.9-2.4 for HOBr+Br⁻, pH = 6.4 for HOBr+Cl⁻ (Beckwith et al., 1996; Liu and Margerum, 2001)). This generated excessive BrO concentrations in comparison to observations. Here we revise the calculation of $k_5^I$ to consider the entire range of aerosol and cloud pH, as recommended by Roberts et al. (2014), resulting in much slower rate. We also adopt a new value for $k_3^I$ from a recent laboratory study (Liu and Abbatt, 2020), updating the upper limit of $3.2\times10^9$ M⁻¹ s⁻¹ previously reported by Liu (2000). Details of these updates are in Table 2. The overall result is to have less efficient heterogeneous recycling and mobilization of bromine radicals in both aerosols and clouds.

Wang et al. (2019) found the heterogeneous reaction of HOCl with Cl⁻ to be the dominant global tropospheric source of $Cl_2$ in GEOS-Chem:

$$HOCl(aq) + Cl^- + H^+ \rightarrow Cl_2 + H_2O \qquad (R6)$$

Here we add competing reactions between HOCl and S(IV):

$$HOCl(aq) + HSO_3^- \rightarrow HCl + HSO_4^- \qquad (R7)$$
$$HOCl(aq) + SO_3^{2-} \rightarrow HCl + SO_4^{2-} \qquad (R8)$$

with reaction rate coefficients $k_7 = 2.8\times10^5$ M$^{-1}$s$^{-1}$ and $k_8 = 7.6\times10^8$ M$^{-1}$s$^{-1}$ from Liu and Abbatt (2020) and Fogelman et al. (1989), respectively. (R7) and (R8) are relatively slow and have minor overall impact on the Cl$_y$ chemistry.

Aerosol aqueous-phase reaction of $N_2O_5$ with Cl$^-$ produces ClNO$_2$ that photolyzes in the daytime to return Cl atoms and NO$_2$. The reaction competes with $N_2O_5$ hydrolysis, with the following first-order loss representation for $N_2O_5$:

$$N_2O_5 \xrightarrow{\varphi Cl^-, (1-\varphi)H_2O} \varphi ClNO_2 + (2-\varphi)NO_3^- + 2(1-\varphi)H^+ \qquad (R9)$$

McDuffie et al. (2018a;b) evaluated different model expressions for the reactive uptake coefficient $\gamma_{N_2O_5}$ and the ClNO$_2$ yield $\varphi$, and recommended lower values than previously used in GEOS-Chem by Wang et al. (2019) to account for the effect of organic coating of particles. We previously implemented this update in Wang et al. (2020) and it is now part of GEOS-Chem version 12.9.

We update the previous GEOS-Chem representation of IBr and ICl formation from uptake of iodine species on seas salt aerosols (Sherwen et al., 2016a) to conserve mass and be consistent with analogous reactions for uptake of bromine and chlorine:

$$INO_3 + 0.85Cl^- + 0.15Br^- + H^+ \rightarrow 0.85ICl + 0.15IBr + HNO_3 \qquad (R10)$$
$$INO_2 + 0.85Cl^- + 0.15Br^- + H^+ \rightarrow 0.85ICl + 0.15IBr + HNO_2 \qquad (R11)$$
$$HOI + 0.85Cl^- + 0.15Br^- + H^+ \rightarrow 0.85ICl + 0.15IBr + H_2O \qquad (R12)$$
$$INO_3 + H_2O \rightarrow HOI + HNO_3 \qquad (R13)$$

Reaction rates are calculated using reactive uptake coefficients $\gamma$ for INO$_3$, INO$_2$, and HOI as given by Sherwen et al. (2016a), with (R10)-(R12) taking place in acidic aerosols and (R13) taking place in alkaline aerosols.

Additional updates to the GEOS-Chem halogen mechanism in version 12.9 include a new scheme to calculate the reactive uptake
coefficients $\gamma$ on ice crystals following recommendations by the International Union of Pure and Applied Chemistry (IUPAC) (Crowley et al., 2010) as listed in Table 3. We calculate the effective radius of ice crystals based on air temperature following Heymsfield et al. (2014) and Holmes et al. (2019), and increase the resulting surface area by a factor of 2.25 to account for irregular shape (Schmitt and Heymsfield, 2005). We also update BrNO$_3$ hydrolysis to include the temperature dependence of $\Gamma$ (in equation 1) from Deiber et al. (2004):

$$\Gamma = 0.0021T - 0.561 \qquad (5)$$

where $T$ is air temperature in K.

# 3 Global budget and distribution of tropospheric halogens

Figure 1 shows the global budgets and cycling of tropospheric inorganic chlorine (1a), bromine (1b), and iodine (1c) in our model simulation. Figure 2 shows the annual mean global distributions of Cl atoms, BrO, and IO. Figure 3 shows the global mean vertical distribution of the halogen speciation for reactive chlorine (Cl* ≡ $Cl_y$ – HCl), $Br_y$, and $I_y$. GEOS-Chem is driven here by 2016 GEOS-FP (forward processing) assimilated meteorological fields from the NASA Global Modeling and Assimilation Office (GMAO) with native horizontal resolution of 0.25° x 0.3125° and 72 vertical levels from the surface to the mesosphere. Our model simulation is conducted at 4° x 5° horizontal resolution and meteorological fields are conservatively degraded to that resolution. The simulation is conducted for two years (2015-2016) with the first year as spin up for initialization.

## 3.1 Chlorine

The dominant global source of $Cl_y$ is acid displacement from SSA to HCl. The global rate of HCl generation from acid displacement is 46 Tg Cl $a^{-1}$ and close to the observationally based estimate of 50 Tg Cl $a^{-1}$ by Graedel and Keene (1995). HCl is the largest reservoir of tropospheric $Cl_y$, with a global mean tropospheric mixing ratio of 45 ppt. Most of HCl is removed by deposition, and only a small fraction (7.3 Tg Cl $a^{-1}$) reacts with OH and contributes to reactive chlorine Cl*. Cl* can be also generated from $Cl^-$ and dissolved HCl in clouds and aerosols by heterogeneous reactions with principal contributions from HOBr+$Cl^-$ (2.6 Tg Cl $a^{-1}$), HOCl+$Cl^-$ (1.5 Tg Cl $a^{-1}$), HOI/$INO_x$ + $Cl^-$ (0.8 Tg Cl $a^{-1}$), and $N_2O_5$+$Cl^-$ (0.68 Tg Cl $a^{-1}$). This heterogeneous source of 6.3 Tg Cl $a^{-1}$ is lower than our previous estimate of 12 Tg Cl $a^{-1}$ in Wang et al. (2019), since the updated mechanisms for HOBr+$Cl^-$ (Section 2.3) and $N_2O_5$+$Cl^-$ (Section 2.4 and Wang et al. (2020)) reactions are slower. We calculate a tropospheric lifetime of 2.3 hours for Cl*. Loss of Cl* is mainly through the reaction of Cl with methane (44%) and other organic compounds.

Distributions of Cl* in the troposphere are generally similar to Wang et al. (2019). As shown in Figure 2, tropospheric Cl atom concentrations are highest at the surface, reflecting the source from SSA (Figure S1), and in the upper troposphere due to transport from the stratosphere as well as cold temperature slowing down the Cl + methane reaction. In surface air, simulated Cl atom concentrations are usually highest along polluted coastlines where the large sources of $HNO_3$, $H_2SO_4$, and $N_2O_5$ from anthropogenic emissions drive acid displacement and $ClNO_2$ production. Figure 3 shows the global mean vertical distribution of Cl* species. Boundary layer Cl* is dominated on a zonal mean basis by $ClNO_2$ formed from $N_2O_5$+$Cl^-$ in polluted air. High mixing ratios of $ClNO_3$ in the upper troposphere are related to transport from the stratosphere and its slow hydrolysis. The BrCl mixing ratio is much lower than in the previous GEOS-Chem studies of Sherwen et al. (2016b) and of Zhu et al. (2019) (who reported a tropospheric mean mixing ratio of 0.69 ppt) because of slower update kinetics of HOBr in aerosol and cloud water.

## 3.2 Bromine

The largest source of $Br_y$ is from SSA debromination in the marine boundary layer (MBL), mainly contributed by HOBr+$Br^-$ and $O_3$+$Br^-$ producing $Br_2$ and HOBr, respectively. Bromocarbon photochemistry dominates the source of $Br_y$ in the free troposphere. Uptake of HBr by SSA is the major sink of $Br_y$. The global tropospheric loading of BrO in the model is 2.1 Gg Br, corresponding to a mean tropospheric mixing ratio of 0.19 ppt (0.38 ppt in daytime). This value is much lower than the most recent GEOS-Chem estimate of 8.0 Gg by Zhu et al. (2019), because of the updated HOBr heterogeneous chemistry described in Section 2.3. The newly added pH-dependences in Table 2 decrease the rate of reaction (R5), resulting in much slower recycling of HOBr in cloud and aerosol water. HOBr is now more likely to react with S(IV) via reactions (R3) and (R4) than previously, forming HBr which then

gets taken up by SSA. In Zhu et al. (2019), 82% of HOBr heterogeneous reactions were with $Br^-$ and $Cl^-$, and only 18% were with S(IV). Due to the update in Section 2.3, 59% of HOBr heterogeneous reactions are with $Br^-$ and $Cl^-$, and 41% are with S(IV). The higher fraction of $Br_y$ as HBr decreases the tropospheric lifetime of $Br_y$ because HBr is more water-soluble than other $Br_y$ species. We calculate tropospheric lifetimes of 7.9 hours for $Br_y$ and 6.8 minutes for $BrO_x$ ($\equiv$ Br+BrO).

Distributions of BrO in Figure 2 are similar to Zhu et al. (2019) except for lower mixing ratios. High surface BrO mixing ratios are usually associated with high SSA (Figure S1). BrO mixing ratios are low over the Southern Ocean despite high SSA emission because SSA alkalinity is not completely depleted and hence reaction (R5) is ineffective. BrO decreases from the surface to the middle troposphere, reflecting the SSA source, and then increases in the upper troposphere because of efficient heterogeneous recycling of HBr in ice clouds (Table 3). Figure 3 shows the global mean vertical distribution of $Br_y$ species, which is very different from Sherwen et al. (2016b) where the $Br_y$ concentration increased with altitude. This is due to the inclusion of SSA debromination in our simulation. Our $Br_y$ mixing ratio in the MBL is still only slightly higher than that in Sherwen et al. (2016b) because of the much lower $Br_y$ lifetime resulting from the slower HOBr heterogeneous reactions, as mentioned above.

### 3.3 Iodine

The $I_y$ source totals 2.7 Tg I $a^{-1}$ with most (2.1 Tg I $a^{-1}$) originating from ocean volatilization of HOI and $I_2$ (Carpenter et al., 2013;MacDonald et al., 2014). The sink of $I_y$ is from deposition (1.8 Tg I $a^{-1}$) and uptake by aerosols (0.91 Tg I $a^{-1}$). The global tropospheric loading of IO in the model is 1.4 Gg I, corresponding to a mean tropospheric mixing ratio of 0.08 ppt. As shown in Figure 2 and 3, concentrations of all $I_y$ species are highest in the MBL, consistent with the dominant emission from the ocean. Surface IO mixing ratios are highest over tropical oceans, where both organic and inorganic iodine emissions are high due to the high temperature. Concentrations of IO and most $I_y$ species are the lowest in middle troposphere where $I_y$ speciation is mostly as HOI, which can be removed via wet deposition efficiently. We calculate tropospheric lifetimes of 1.6 days for $I_y$ and 1.7 minutes for I+IO* ($\equiv$ IO+OIO+$2I_2O_2$+$2I_2O_3$+$2I_2O_4$). Our results are consistent with Sherwen et al. (2016b) since the iodine chemistry is largely unchanged. Our only significant update has been to conserve mass in iodine heterogeneous reactions (Text S2) but this has little impact.

## 4 Comparison to observations

Here we compare the model simulation for 2016 to observations for gas-phase halogen species collected from surface and aircraft campaigns. The observations are in different years but we assume that interannual variability is small compared to other sources of error. More extensive evaluations of previous model versions with observations for organohalogens, HCl/$Cl^-$ acid displacement, and iodine species can be found in Sherwen et al. (2016b), Wang et al. (2019), and Sherwen et al. (2016a) respectively, and our model results are not significantly different for purpose of these comparisons.

### 4.1 Bromine enrichment factors (EF)

The bromine enrichment factor (EF) is a measure of SSA debromination, which can be calculated in the model as:

$$EF = \frac{([Br^-]/[SSA])_{atmosphere}}{([Br^-]/[SSA])_{emission}} \qquad (6)$$

Figure 4 shows the annual mean EFs in surface air in GEOS-Chem. The high values (EF > 1) indicate a more important role of HBr uptake than SSA debromination. EF is especially high over continental regions because $Br_y$ volatilized from SSA is then transported inland and taken up by continental aerosols. Measured annual mean observations at 10 surface sites from Sander et al. (2003) and from Newberg et al. (2005) are also shown in Figure 4. The mean GEOS-Chem EF averaged over these sites is 0.88, higher that in Zhu et al. (2019) (0.75). This is due to the updated reactive uptake of HOBr in Section 2.2, which results in less efficient mobilization of bromine radicals from SSA. The mean observed EF is 0.57. The model bias is mainly due to the underestimates over the Southern Ocean. Zhu et al. (2019) suggested that this may be due to excessive model uptake of HBr by SSA in summer. Free tropospheric transport of bromine released from SSA (Wang et al. 2015) is estimated conservatively in GEOS-Chem, as the updated HOBr reactive uptake may potentially lead to overestimate of bromine wash out during deep convection.

## 4.2 Bromine monoxide (BrO)

Figure 5 compares surface measurements of BrO concentrations in marine air during daytime with corresponding model values. The model is generally consistent with these observations in showing surface air BrO mixing ratios in the range 0–3 ppt. BrO over the tropical North Atlantic is higher (1-3 ppt) than other oceans (<1 ppt and below measurement detection limits) in both the model and observations. This is due in the model to high SSA emissions and efficient acidification of SSA from continental outflow of $HNO_3$ and $SO_2$, resulting in rapid debromination. Figure 6 compares modeled vertical profiles with aircraft BrO observations over the tropics from the CONTRAST (Chen et al., 2016; Koenig et al., 2017), CAST (Le Breton et al., 2017), TORERO (Volkamer et al., 2015), and ATom (Wofsy et al., 2018; Veres et al., 2019) aircraft campaigns. Details of the instrument and uncertainty of these observations are listed in Table 4. The median profiles of BrO measured by CIMS during CONTRAST, CAST, and ATom are all around or below their detection limits. In contrast, observations during CONTRAST and TORERO measured by DOAS show higher BrO mixing ratios (> 1 ppt). There are two independent BrO measurements during CONTRAST. The DOAS measurement by Koenig et al. (2017) are portions of five flights during CONTRAST, and show higher values than the CIMS measurement by Chen et al. (2016). The model provides a reasonable fit to CONTRAST CIMS BrO with mean bias of 0.03 ppt, but is low compared to the DOAS observations. Observed BrO mixing ratios are low almost everywhere during the ATom campaign and show no obvious vertical variation from MBL to free troposphere. Modeled BrO is generally consistent with ATom observations in the lower troposphere but is much higher in the upper troposphere where transport from the stratosphere becomes important in the model. On the other hand, the model is lower than the TORERO observations in the upper troposphere. The higher BrO mixing ratios in the lowermost stratosphere in the model during ATom and in both model and observations during TORERO are consistent with balloon-borne measurements at 45°N by Stachnik et al. (2013) showing 5 ppt BrO at 15km altitude, but the lower BrO mixing ratios in the observations during ATom and in both model and observations during CONTRAST CIMS are consistent with aircraft measurements over the eastern Pacific by Werner et al. (2017) showing < 1 ppt at 12-15km altitudes.

To summarize, there is much ambiguity in the comparisons of model results to observed BrO concentrations, as might be expected since most observations are near their detection limits and with large uncertainties (Table 4). There is no evidence of systematic model bias but more sensitive observations would be needed to be conclusive.

### 4.3 Inorganic chlorine gases (Cl$_y$)

Our model does not include anthropogenic inorganic chlorine sources, which could however be important in polluted continental boundary layer regions as seen in atmospheric observations (Wang et al., 2016; Tham et al., 2016; Lee et al., 2018; Zhou et al.,2018; Yun et al., 2018; Peng et al., 2020; Thornton et al., 2020; Wang et al., 2020; Gunthe et al., 2021). Here we focus on a more global perspective. Figure 7 compares modeled surface HCl mixing ratios to observations at coastal sites and over oceans. The model captures the spatial variability of the HCl mixing ratios across locations, which largely reflects the strong acid displacement at northern midlatitudes. As previously shown by Wang et al. (2019), acid displacement is key to reproducing the observations. Figure 8 compares surface modeled maximum ClNO$_2$ to observations at island and coastal environments. Observations of ClNO$_2$ are usually reported as maxima instead of means, and are made in nighttime urban environments, which are difficult to compare to our global model because of the coarse grid resolution and nighttime stratification of the surface layer. Despite these drawbacks, the model still offers a credible simulation of the 24-h maximum ClNO$_2$.

The WINTER aircraft campaign provided data for multiple Cl$_y$ gases including HCl, ClNO$_2$, HOCl, and Cl$_2$. The measurements were made over the eastern US and offshore during February–March 2015 by I-TOF-CIMS (Lee et al., 2018), as summarized in Table 4. Figure 9 compares the observed median vertical profiles of HCl, ClNO$_2$, HOCl, and Cl$_2$ during WINTER to the model sampled along the flight tracks for the corresponding period. Modeled HCl is lower than the observations but mostly within the calibration uncertainty ($\pm$ 30%). Modeled HOCl largely underestimates WINTER observations. Wang et al. (2019) found that such underestimation is over both land and ocean and mainly in daytime when HOCl has very short lifetime against photolysis (a few minutes). This may suggest a large photochemical source needed to decrease the model bias. Recent work also identified to the potential for IOx- ion chemistry to lead to measurement interferences (Dörich et al., 2021), of the detection of acid gases which could impact the measured HOCl:HCl ratio. Furthermore, rapid interconversion of halogen species on inlet walls have been reported that could also impact the measured HOCl:HCl ratio (Neuman et al., 2010).

Figure 10 compares modelled vertical BrCl, Cl$_2$, and ClNO$_2$ mixing ratios to observations during the ATom aircraft campaigns. Both modeled and observed chlorine gases are low in most regions (< 1ppt). Most ATom measurements were made in daytime, when modeled BrCl, Cl$_2$, and ClNO$_2$ are close to zero due to their very short lifetimes against photolysis. Modeled BrCl and Cl$_2$ underestimate observed values especially in lower troposphere. The observed median mixing ratios of all these species at all altitudes are either below or around the measurement detection limits (Table 4). The underestimates of HOCl during WINTER, and BrCl, Cl$_2$ during ATom at daytime may suggest a large photochemical source that can produce chlorine radicals from Cl$^-$.

### 4.4 Iodine monoxide (IO)

Figure 11 compares surface measurements of IO over islands and oceans during daytime with corresponding model values. The model is generally consistent with these measurements with an overall bias of -10%. Both modeled and observed IO mixing ratios are highest over tropical oceans and lowest at high latitudes, reflecting the distribution of both organic and inorganic iodine emissions. Figure 12 compares modeled vertical profiles with aircraft IO observations over the eastern Pacific from TORERO (Volkamer et al., 2015). The model is in general agreement with the observations and able to reproduce the observed vertical variation with a mean bias of -0.09ppt. Both observed and modeled IO mixing ratios are high in MBL, reflecting the marine sources of iodine, and vary little in the free troposphere. Recently, Koenig et al. (2020) reported IO and I$_y$ mixing ratios of 0.08 and 0.53

ppt at 12 km during the CONTRAST campaign over western tropical Pacific. Our modeled values are 0.07 and 0.43 ppt for IO and $I_y$ respectively at that location.

## 5 Global implications for tropospheric oxidant chemistry

We now examine the implications of tropospheric halogen chemistry as described by our mechanism on the concentrations of tropospheric VOCs, ozone, $NO_x$, and OH. Shah et al. (2021) examined the implications for mercury chemistry.

### 5.1 Volatile organic compounds (VOCs)

Cl atoms are strong VOC oxidants, but their importance is limited by their small supply. The global mean tropospheric Cl atom concentration in our model is 630 cm$^{-3}$, consistent with the upper limit of 1000 cm$^{-3}$ inferred by Singh et al. (1996) from global modeling of $C_2Cl_4$ observations. Within the MBL, the global mean concentration is 840 cm$^{-3}$. similar to a recent estimate using isotopic observations of methane and CO by Gromov et al. (2018) (900 cm$^{-3}$). Oxidation by Cl atoms in troposphere drives a loss rate of 3.6 Tg a$^{-1}$ for methane in our model, contributing 0.8% of the total methane chemical loss. It additionally contributes 14% of the global chemical loss for ethane, 8% for propane, and 7% for higher alkanes. These impacts could be higher if anthropogenic chlorine sources were considered. Oxidation of VOCs by Br atoms in GEOS-Chem is significant only for acetaldehyde, where it accounts for 2.0% of the global loss, and up to 18% of the loss in the MBL of high-SSA regions (tropical oceans, North Atlantic). Badia et al. (2019) previously estimated a 9% contribution of Br atoms to acetaldehyde oxidation in the tropospheric column over the eastern tropical Pacific.

### 5.2 Ozone, $NO_x$, and OH

Figure 13 shows the effects of halogen chemistry on tropospheric OH, $NO_x$, and ozone concentrations, as obtained by difference with a sensitivity simulation excluding all halogen reactions in the troposphere ("no halogen"). Halogen chemistry decreases the global tropospheric ozone burden by 11% in our model, which is smaller than the 18.6% in Sherwen et al. (2016b). Global ozone chemical production decreases by 2% while ozone lifetime decreases by 10%. The decrease in ozone production is due to a 5.6% global decrease in $NO_x$ as a result of formation and hydrolysis of halogen nitrates $X$NO$_3$ ($X \equiv$ Cl, Br, I):

$$X\text{O} + \text{NO}_2 + \text{M} \rightarrow X\text{NO}_3 + \text{M} \qquad \text{(R21)}$$
$$X\text{NO}_3 + \text{H}_2\text{O} \rightarrow \text{HO}X + \text{HNO}_3 \qquad \text{(R22)}$$

Globally, such $NO_x$ loss is mostly through ClNO$_3$ and BrNO$_3$ hydrolysis, with negligible contribution from INO$_3$. As shown in Figure 13, surface $NO_x$ increases over the continents and this is due to ClNO$_2$ chemistry. We previously showed in Wang et al. (2019) that Cl$^-$ originating from SSA can be transported far inland by acid displacement of HCl and subsequent HCl uptake by sulfate-nitrate-ammonium (SNA) aerosols. Cl$^-$ will then react with N$_2$O$_5$ over the continents via reaction (R9) and form ClNO$_2$, resulting in longer $NO_x$ lifetime. This increase in continental boundary layer $NO_x$ would be further amplified by anthropogenic sources of Cl$^-$. Halogen chemistry in our model lowers global tropospheric concentrations of OH and HO$_2$ by 4.1% and 3.4% respectively. Decrease in OH is mainly due to the decrease of ozone, which reduces primary OH production from ozone by 9.8%. The increase in OH over continental regions (Figure 13) is due to ClNO$_2$ chemistry.

Table 5 summarizes the global annual budget of tropospheric ozone in the standard model and in the no halogen simulation. The budget of ozone is shown as that of odd oxygen ($O_x \equiv O_3 + O + O(^1D) + NO_2 + 2NO_3$ + peroxyacylnitrates + $HNO_3$ + $HNO_4$ + $3N_2O_5$ + organic nitrates + Criegee intermediates + $XO$ + $HOX$ + $XNO_2$ + $2XNO_3$ + $2OIO$ + $2I_2O_2$ + $3I_2O_3$ + $4I_2O_4$ + $2Cl_2O_2$ + $2OClO$ where $X \equiv$ Cl, Br, I) to account for the rapid cycling between $O_x$ species. The 10% shorter ozone lifetime as a result of halogen chemistry is due to catalytic ozone loss cycles driven by iodine (7.6%), bromine (2.6%) and chlorine (0.3%). Figure 14 shows the relative contributions of different reaction routes to ozone chemical loss in troposphere. Halogens contribute about 19% of ozone loss in the MBL, decreasing to 8% at 2-4 km altitude and then increasing to 24% in the upper troposphere. Halogen-catalyzed ozone loss is mainly driven by the sequence ($X \equiv$ I, Br, Cl):

$$X + O_3 \rightarrow XO + O_2 \qquad\qquad (R23)$$
$$XO + HO_2 \rightarrow HOX + O_2 \qquad\qquad (R24)$$
$$HOX + hv \rightarrow X + OH \qquad\qquad (R25)$$

Bates and Jacob (2020) introduced an expanded odd oxygen family, $O_y \equiv O_x + O_z$, to include both $O_x$ and an additional subfamily, $O_z$, consisting of $HO_x$ and its reservoirs ($O_z \equiv 0.5 \times$(H + OH + organic peroxy radicals + $HNO_2$ + $HNO_3$ + $HNO_4$ + peroxyacylnitrates + organic nitrates + X + XO + $XNO_2$ + $XNO_3$ + OIO + OClO + ClOO) + $H_2O_2$ + organic peroxides + $X_2$ + HOX + $I_2O_2$ + $I_2O_3$ + $I_2O_4$ + $Cl_2O_2$ where X $\equiv$ Cl, Br, I). Table 4 also summarizes the budget of $O_z$. The global tropospheric $O_z$ burden decreases by 4% due to the halogen chemistry, which is mainly because of the lower production from $O_x$. Following Bates and Jacob (2020), we define the chain length $N$, or $O_x$ production efficiency per unit $O_z$, as the number of times a unit of $O_z$ is converted to $O_x$ before it is removed to terminal sinks:

$$N = \frac{\text{Rate of conversion from } O_z \text{ to } O_x}{(\text{Rate of } O_z \text{ loss to } H_2O) + (\text{Rate of } O_z \text{ deposition})} \qquad\qquad (7)$$

In the conventional $O_x$ budget analysis, conversion from $O_x$ to $O_z$ through $O(^1D) + H_2O$, is viewed as a sink for $O_x$; but if $N > 1$ it is actually a net source. By considering this, Bates and Jacob (2020) introduced an effective ozone lifetime as:

$$\tau = \frac{1}{k_{Ox \text{ loss to } O_2} + k_{Ox \text{ deposition}} + (1-N)k_{Ox \text{ conversion to } Oz}} \qquad\qquad (8)$$

where $k_i$ is the pseudofirst-order loss rate constant for process $i$. As shown in Table 4, $N$ increases from 1.40 to 1.47 by including halogen chemistry, thus amplifying ozone production efficiency from $O(^1D) + H_2O$. This is because of the decrease of $HO_2$ which slows down the loss rate of $HO_x$. The effective ozone lifetime decreases by 15%, from 71 to 60 days, because the halogen-driven catalytic pathways represent true ozone sinks by converting $O_3$ to $O_2$.

Figure 15 compares modeled ozone concentrations with and without halogen chemistry to ozonesonde observations from the World Ozone and Ultraviolet Data Center (WOUDC, http://www.woudc.org). We only use data from Electrochemical Concentration Cell (ECC) and do not apply WOUDC-suggested correction factors, following Hu et al. (2017). There are a total of 47 stations in 2016 (supplemental Table S1) and we average the data into six latitudinal bands. Halogen chemistry does not degrade the simulation in the Southern Hemisphere, where the model bias is small, but worsens the underestimate in the Northern Hemisphere. Similar results are found in Figure 16, which compares modeled surface ozone mixing ratios to observations at surface sites. There is no significant seasonal variation for the impacts of halogen chemistry on surface ozone at these sites. The last extensive evaluation of

the global tropospheric ozone simulation in GEOS-Chem was done by Hu et al. (2017) and found no significant bias, but it used v10.1 of the model and there have been many changes to the model since then. In particular, the introduction of $NO_y$ reactive uptake by clouds in version 12.6 (Holmes et al., 2019) drove a 7% decrease in global tropospheric ozone. Correcting this underestimate should be a topic of further research.

**6 Conclusions**

We presented a new comprehensive representation of tropospheric halogen chemistry in the GEOS-Chem model that synthesizes and updates previous model developments. We used it to analyze the sources and cycling of halogen radicals, evaluate against observations of halogen radicals and their reservoirs, and examine the implications for tropospheric oxidant chemistry.

The model includes an improved representation of heterogeneous chemistry in aerosols and clouds, including in particular the reactions of HOBr, leading to less effective recycling and mobilization of bromine radicals. This allows us to include in the model the known source of bromine radicals from debromination of sea salt aerosols (SSA) without generating excessive BrO concentrations. Simulation of cloud processing is improved to include a more accurate computation of cloudwater pH (Shah et al., 2020) and cloud entrainment (Holmes et al., 2019). $ClNO_2$ production by the heterogeneous $N_2O_5 + Cl^-$ reaction is updated to a slower rate to account for organic coating of particles (McDuffie et al., 2018a;b).

Cycling of chlorine and iodine radicals is similar to previous versions of GEOS-Chem (Wang et al., 2019; Sherwen et al., 2016) but cycling of bromine radicals is very different. We find a mean tropospheric BrO mixing ratio of 0.19 ppt, much lower than previous GEOS-Chem estimates and reflecting the less effective heterogeneous recycling of bromine radicals. BrO is highest in the marine boundary layer (MBL) where SSA debromination is the main source, and in the upper troposphere, due to photodecomposition of bromocarbons and transport from the stratosphere. Model results are consistent with MBL observations of BrO from coastal sites and ship cruises, though observations are often below the detection limit. Comparisons to vertical profiles from aircraft campaigns paints an inconsistent picture, with model BrO being lower than the CAST CIMS, CONTRAST DOAS, and TORERO DOAS measurements over the tropical Pacific, but higher than the ATom CIMS measurements at high altitudes on Pacific and Atlantic transects. The TORERO and CONTRAST DOAS data show increasing BrO concentrations in the upper troposphere but the ATom CIMS data do not. The aircraft observations are again below or close to detection limits. A more confident evaluation of tropospheric bromine radical chemistry will require more sensitive observations of BrO and its reservoirs in the future.

Our simulation shows a global mass-weighted mean Cl atom concentration of 630 molecules $cm^{-3}$ in the troposphere. Oxidation by Cl atoms accounts for 0.8% of the global loss of atmospheric methane and has larger effects on the global losses of ethane (14%), propane (8%), and higher alkanes (7%). Reactive chlorine ($Cl^* \equiv Cl_y - HCl$) is mainly generated from HCl + OH (7.3 Tg Cl $a^{-1}$), heterogeneous reactions of $Cl^-$ in clouds (6 Tg Cl $a^{-1}$) and oxidation of organochlorines (3.3 Tg Cl $a^{-1}$). Comparisons of model results to observations in marine surface air and aircraft campaigns in this study and our previous work (Wang et al., 2019) show that the model is in general consistent with the range and distributions of observed HCl and $ClNO_2$ concentrations. The model cannot reproduce the high daytime BrCl, and $Cl_2$ concentrations observed during ATom, and matching those values would require a fast $Cl^*$ source. Whether this can be compatible with other ATom observations of VOCs and radicals needs future investigation.

Our simulated IO mixing ratios are consistent with marine observations in surface air and from aircraft, showing high values in the tropical MBL and low uniform values in the free troposphere. Our simulated global mean tropospheric IO concentration is 0.08 ppt.


Halogen chemistry decreases the global burden of tropospheric ozone in GEOS-Chem by 11%. This reflects a 2% decrease in ozone production due to the sink of $NO_x$ from formation and hydrolysis of $ClNO_3$ and $BrNO_3$, and a 11% increase in ozone chemical loss due to catalytic cycles involving iodine (8%) and bromine (3%). The global mean tropospheric OH concentration decreases by 4.1%, mostly due to the decrease in ozone. Tropospheric ozone concentrations in GEOS-Chem show no significant

bias in the Southern Hemisphere relative to ozonesonde data, but a low bias in the Northern Hemisphere that is also present in the absence of halogen chemistry. Addressing this low bias should be a priority for future research.

**Data availability.** The model code is available at GEOS-Chem repository (http://www.geos-chem.org, doi:10.5281/zenodo.3950327). ATom data are publicly available through the Oak Ridge National Laboratory DAAC (https://daac.ornl.gov/ATOM/campaign/data.html). Data from the WINTER, CONTRAST, and TORERO campaigns are publicly

available at the EOL data archive (https://data.eol.ucar.edu/). Data of CAST are publicly available at the CEDA archive (https://catalogue.ceda.ac.uk/uuid/565b6bb5a0535b438ad2fae4c852e1b3). All links mentioned here were last accessed on 1 April 2021.

**Author contributions.** XW and DJJ designed the study and prepared the paper with input from all co-authors. XW developed the updated halogen code, performed the simulations, and conducted the analysis. WD merged the halogen code with other updates of

GEOS-Chem in version 12.9. XW, SZ, LZ, VS, CDH, TS, BA, MJE, and SDE contributed to the GEOS-Chem model development. JAN, PV, TKK, RV, LGH, ML, TJB, CJP, BHL, and JAT conducted and processed the aircraft halogen measurements.

**Competing interests.** The authors declare that they have no conflict of interest.

**Acknowledgments.** This work was supported by the City University of Hong Kong New Research Initiatives (grant no. 9610470) and National Natural Science Foundation of China (grant no. 42005083), and was partially supported by the Shenzhen Research

Institute, City University of Hong Kong. Work at Harvard was supported by the EPA STAR Program (grant no. 84001401). The authors thank Michael Le Breton for CAST measurements and Kelvin H. Bates for helpful discussions.

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

**Table 1. Global sources and sinks of tropospheric gas-phase phase inorganic chlorine (Cl$_y$), bromine (Br$_y$), and iodine (I$_y$)[a].**

| | Cl$_y$ (Tg Cl a$^{-1}$) | Br$_y$ (Tg Br a$^{-1}$) | I$_y$ (Tg I a$^{-1}$) |
|---|---|---|---|
| Total source | 54 | 21 | 2.7 |
| Sea Salt | 50 | 20 | - |
| Acid displacement [b] | 46 | - | - |
| HO$X$ + $X^-$ | 2.4 | 12 | - |
| $X$NO$_2$ + $X^-$ | 0.05 | < 0.01 | - |
| OH/O$_3$ + $X^-$ | 0.53 | 8.6 | - |
| $X$NO$_3$ + $X^-$ | 0.15 | 0.11 | - |
| N$_2$O$_5$ + Cl$^-$ | 0.68 | - | - |
| HOI and I$_2$ ocean emission | NA | NA | 2.1 |
| Organohalogens [c] | 3.3 | 0.54 | 0.58 |
| CH$_3X$ + OH/$hv$ | 2.0 | 0.05 | 0.26 |
| CH$_2X_2$ +OH/$hv$ | 0.88 | 0.06 | 0.11 |
| CH$X_3$ + OH/$hv$ | 0.30 | 0.40 | - |
| CH$_2$I$X$ + OH/$hv$ | 0.04 | 0.03 | 0.21 |
| Stratosphere [d] | 0.14 | 0.01 | < 0.01 |
| Open fires | 0.50 | - | - |
| Total sink | 54 | 21 | 2.7 |
| Deposition | 54 | 1.4 | 1.8 |
| Dry | 29 | 0.70 | 0.93 |
| Wet | 25 | 0.70 | 0.84 |
| Net uptake by aerosols | NA [e] | 20 | 0.91 |
| Tropospheric mass (Gg) | 231 | 19 | 12 |
| Lifetime (hours) | 38 | 7.9 | 39 |

[a] Annual totals for 2016 computed from GEOS-Chem. Dashes indicate negligibly small terms. Gas-phase inorganic chlorine is defined as Cl$_y$ ≡ Cl + 2×Cl$_2$ + 2×Cl$_2$O$_2$ + ClNO$_2$ + ClNO$_3$ + ClO + ClOO + OClO + BrCl + ICl + HOCl + HCl. Gas-phase inorganic bromine is defined as Br$_y$ ≡ Br + 2×Br$_2$ + BrNO$_2$ + BrNO$_3$ + BrO + BrCl + IBr + HOBr + HBr. Gas-phase inorganic iodine is defined as I$_y$ ≡ I + 2×I$_2$ + 2×I$_2$O$_2$ + 2×I$_2$O$_3$ + 2×I$_2$O$_4$ + OIO + INO + INO$_2$ + INO$_3$ + IO + ICl + IBr + HOI + HI. We use $X$ to denote any of Cl, Br, or I.
[b] Acid displacement, significant only for HCl. The Table gives the net production minus loss of HCl from acid aerosol displacement by HNO$_3$ and H$_2$SO$_4$, minus HCl uptake by sea salt alkalinity.
[c] CH$_2X_2$ and CH$X_3$ denote unmixed halogens, such as CH$_2$Cl$_2$ or CHBr$_3$.CH$_2$I$X$ denotes the mixed iodocarbons CH$_2$ICl and CH$_2$IBr.
[d] Net stratospheric input to the troposphere.
[e] For Cl$_y$ the uptake is included as an offsetting term in the acid displacement source (footnote b).

**Table 2. First-order reaction rate constants ($k^I$) for HOBr heterogeneous reactions in aerosol and liquid cloud water.**

| | Reaction | First-order reaction rate constant ($k^I$) [a] | Reference [b] |
|---|---|---|---|
| R3 | $HOBr + HSO_3^- \rightarrow HBr + HSO_4^-$ | $k^I = k^{II}[HSO_3^-]$ <br><br> $k^{II} = 2.6 \times 10^7$ M$^{-1}$s$^{-1}$ | (1) |
| R4 | $HOBr + SO_3^{2-} \rightarrow HBr + SO_4^{2-}$ | $k^I = k^{II}[SO_3^{2-}]$ <br><br> $k^{II} = 5 \times 10^9$ M$^{-1}$s$^{-1}$ | (2) |
| R5 | $HOBr(aq) + YBr^- + (1-Y)Cl^- + H^+$ <br> $\rightarrow YBr_2 + (1-Y)BrCl + H_2O$ | $k^I = k_1^{II}[Br^-] + k_2^{II}[Cl^-]$ <br><br> $Y = 0.41\log_{10}\left(\dfrac{[Br^-]}{[Cl^-]}\right) + 2.25$ for $[Br^-]/[Cl^-] < 5 \times 10^{-4}$ <br><br> $Y = 0.90$ for $[Br^-]/[Cl^-] \geq 5 \times 10^{-4}$ | (3) |
| | | $k_1^{II} = \begin{cases} 1.6 \times 10^8 \text{ M}^{-1}\text{s}^{-1} \text{ for pH} \leq 2 \\ k_{ter}[H^+] \text{ for } 2 > \text{pH} < 6 \\ 1.6 \times 10^4 \text{ M}^{-1}\text{s}^{-1} \text{ for pH} \geq 6 \end{cases}$ <br><br> $k_{ter} = 1.6 \times 10^{10}$ M$^{-2}$s$^{-1}$ | (4) |
| | | $k_2^{II} = \begin{cases} 2.3 \times 10^4 \text{ M}^{-1}\text{s}^{-1} \text{ for pH} \leq 6 \\ k_{ter}[H^+] \text{ for pH} > 6 \text{ and pH} < 9 \\ 23 \text{ M}^{-1}\text{s}^{-1} \text{ for pH} \geq 9 \end{cases}$ <br><br> $k_{ter} = 2.3 \times 10^{10}$ M$^{-2}$s$^{-1}$ | (5) |

[a] This first-order rate constant describes the first-order HOBr loss rate $-\dfrac{d[\mathbf{HOBr}]}{dt} = k^I [\mathbf{HOBr}]$ which is used in equations (1)-(4) to calculate the HOBr reactive uptake coefficient $\gamma$.

[b] References: (1) Liu and Abbatt (2020); (2) Troy and Margerum (1991); (3) Fickert et al. (1999); (4) Roberts et al. (2014), $k_{ter}$ value is from Beckwith et al. (1996); (5) Roberts et al. (2014), $k_{ter}$ value is from Liu and Margerum (2001).

**Table 3. Heterogeneous halogen reactions on ice crystals [a]**

| | Reaction | Reactive uptake coefficient ($\gamma$) |
|---|---|---|
| R14 | $HOBr + HCl \rightarrow BrCl + H_2O$ | $\gamma = \gamma_{gs}\theta(HCl)$; $\gamma_{gs} = 0.25$ <br><br> $\theta(HCl) = \frac{K_{LangC,HCl}[HCl]_g}{1+K_{LangC,HCl}[HCl]_g+K_{LangC,HNO3}[HNO_3]_g}$ ; <br><br> $K_{LangC,HCl} = \frac{K_{LinC,HCl}}{N_{max,HCl}}$; $K_{LangC,HNO3} = \frac{K_{LinC,HNO3}}{N_{max,HNO3}}$ <br><br> $K_{LinC,HCl} = 1.3 \times 10^{-5} e^{(4600/T)}$; $N_{max,HCl} = 3 \times 10^{14}$ molecules cm$^{-2}$ <br><br> $K_{LinC,HNO3} = 7.5 \times 10^{-5} e^{(4585/T)}$; $N_{max,HNO3} = 2.7 \times 10^{14}$ molecules cm$^{-2}$ |
| R15 | $HOBr + HBr \rightarrow Br_2 + H_2O$ | $\gamma = \gamma_{gs}\theta(HBr)$; $\gamma_{gs} = 4.8 \times 10^{-4} e^{(1240/T)}$ <br><br> $\theta(HBr) = 4.14 \times 10^{-10}[HBr]_g^{0.88}$ |
| R16 | $HOCl + HCl \rightarrow Cl_2 + H_2O$ | $\gamma = \gamma_{gs}\theta(HCl)$ ; $\gamma_{gs} = 0.22$ |
| R17 | $BrNO_3 + H_2O \rightarrow HOBr + HNO_3$ | $\gamma = 5.3 \times 10^{-4} e^{(1100/T)}$ |
| R18 | $ClNO_3 + H_2O \rightarrow HOCl + HNO_3$ | $\gamma = 1/(\frac{1}{\alpha_s} + \frac{c}{4k_s K_{linC,ClNO3}[H_2O]_s})$ <br><br> $\alpha_s = 0.5$; $k_s K_{linC,ClNO3} = 5.2 \times 10^{-17} e^{(2032/T)}$ <br><br> $[H_2O]_s = 10^{-15} - 3N_{max,HNO3}\theta(HNO_3)$ <br><br> $\theta(HNO_3) = \frac{K_{LangC,HNO3}[HNO_3]_g}{1 + K_{LangC,HCl}[HCl]_g + K_{LangC,HNO3}[HNO_3]_g}$ |
| R19 | $ClNO_3 + HCl \rightarrow Cl_2 + HNO_3$ | $\gamma = \gamma_{gs}\theta(HCl)$ ; $\gamma_{gs} = 0.24$ |
| R20 | $ClNO_3 + HBr \rightarrow BrCl + HNO_3$ | $\gamma = \gamma_{gs}\theta(HCl)$ ; $\gamma_{gs} = 0.56$ |

[a] Formulations for the reactive uptake coefficient $\gamma$ are from IUPAC (Cowley et al., 2010). [ ]$_g$ denotes gas-phase concentration in unit of molecules per cm$^3$ of air. $\gamma_{gs}$ is the elementary reaction probability for a gas phase molecule colliding with the ice surface. $\theta$ is the fractional coverage of a gas species on the ice surface. $K_{LangC}$ is a partition coefficient in units of cm$^3$ molecule$^{-1}$. $K_{LinC}$ is a partition coefficient in units of molecule cm$^{-2}$/molecule cm$^{-3}$. $T$ is air temperature in K. $N_{max}$ denotes the maximum number of available surface sites for a gas species per cm$^2$ of ice surface. $c$ is the average gas-phase thermal velocity for the reactant. $\theta$, $K_{LangC}$, $K_{LinC}$, and $N_{max}$ for each species are calculated using the same method throughout the table. R14 and R15 compete with each other; R18, R19, and R20 compete with each other; these competitions use branching ratios determined by the relative rates.

**Table 4: Summary of aircraft measurements.**

| Campaign | Location | Time | Instrument | Species | Detection limit | Accuracy | Reference [f] |
|---|---|---|---|---|---|---|---|
| CONTRAST | W. tropical Pacific | Jan-Feb 2014 | CIMS [a] | BrO | 1.0 ppt [c] | 23% | (1) |
| | | | DOAS [b] | BrO | 0.5 ppt [d] | 30% | (2) |
| CAST | W. tropical Pacific | Jan-Feb 2014 | CIMS [a] | BrO | 0.1 ppt [d] | 15% | (3) |
| TORERO | E. tropical Pacific | Jan-Feb 2012 | DOAS [b] | BrO | 0.5 ppt [d] | 30% | (4) |
| | | | | IO | 0.05 ppt [d] | 20% | |
| ATom | Pacific and Atlantic | Sep-Oct 2017 (ATom-3) Apr-May 2018 (ATom-4) | CIMS [a] | BrO | 0.3 ppt [e] | 25% + 0.2 ppt | (5) |
| | | | | BrCl | 0.3 ppt [e] | 25% + 0.4 ppt | |
| | | | | $Cl_2$ | 0.4 ppt [e] | 15% + 0.4 ppt | |
| | | | | $ClNO_2$ | 0.1 ppt [e] | 15% + 0.05 ppt | |
| WINTER | E. US and offshore | Feb-Mar 2015 | CIMS [a] | HCl | 100 ppt [e] | 30% | (6) |
| | | | | $ClNO_2$ | 2 ppt [e] | 30% | |
| | | | | HOCl | 2 ppt [e] | 30% | |
| | | | | $Cl_2$ | 1 ppt [e] | 30% | |

[a] CIMS: Chemical Ionization Mass Spectrometer
[b] DOAS: Differential Optical Absorption Spectroscopy
[c] for 60 seconds data.
[d] for 30 seconds data.
[e] for 1 second data.
[f] References: (1) Chen et al. (2016); (2) Koenig et al. (2017); (3)Le Breton et al. (2017); (4)Dix et al. (2016); (5)Veres et al. (2019); (6) Lee et al. (2018).

**Table 5: Global tropospheric ozone budget in GEOS-Chem[a]**

|  | Version 12.9 [b] | No halogen [c] |
|---|---|---|
| $O_x$ |  |  |
| Sources (Tg a$^{-1}$) |  |  |
|    Chemistry | 4359 | 4450 |
|    Stratosphere | 554 | 543 |
| Sinks (Tg a$^{-1}$) |  |  |
|    Chemistry | 4077 | 4078 |
|      $O(^1D) + H_2O$ | 1960 | 2173 |
|      $O_3 + HO_2$ | 1020 | 1188 |
|      $O_3 + OH$ | 468 | 543 |
|      Bromine | 105 | 0 |
|      Iodine | 310 | 0 |
|      Chlorine | 13 | 0 |
|      Others | 201 | 174 |
|    Deposition | 836 | 915 |
| Tropospheric burden (Tg) | 314 | 353 |
| Lifetime (days) | 23.4 | 26 |
| $O_z$ |  |  |
| Sources (Tg a$^{-1}$) |  |  |
|    $O_x \rightarrow O_z$ | 2042 | 2264 |
|    Carbonyl photolysis | 931 | 912 |
| Sinks (Tg a$^{-1}$) |  |  |
|    $O_z \rightarrow H_2O$ | 2361 | 2515 |
|    Deposition | 611 | 661 |
| Tropospheric burden (Tg) | 7.1 | 7.4 |
| Chain length $N$ [d] | 1.47 | 1.40 |
| Effective ozone lifetime (days) [e] | 60 | 71 |

[a] Annual mean budget for the odd oxygen family ($O_x$) and for the reservoirs ($O_z$) of the expanded odd oxygen family ($O_y \equiv O_x + O_z$). Here, $O_z \approx$ 0.5 $HO_y$ accounts for the hydrogen oxide ($HO_x \equiv OH$ + peroxy radicals) and their reservoirs cycling with ozone. See the text in Section 5.2 and Bates and Jacob (2020) for details. All values are given in ozone equivalent mass. For the halogen crossover reactions where two different halogens are included (e.g. $ClO + BrO$), we split the ozone loss equally between the two halogens.
[b] as implemented in this work.
[c] version 12.9 with no tropospheric halogen reactions.
[d] $O_x$ production efficiency per unit $O_z$, see equation (6) in the text for definition.
[e] See equation (7) in the text for definition.

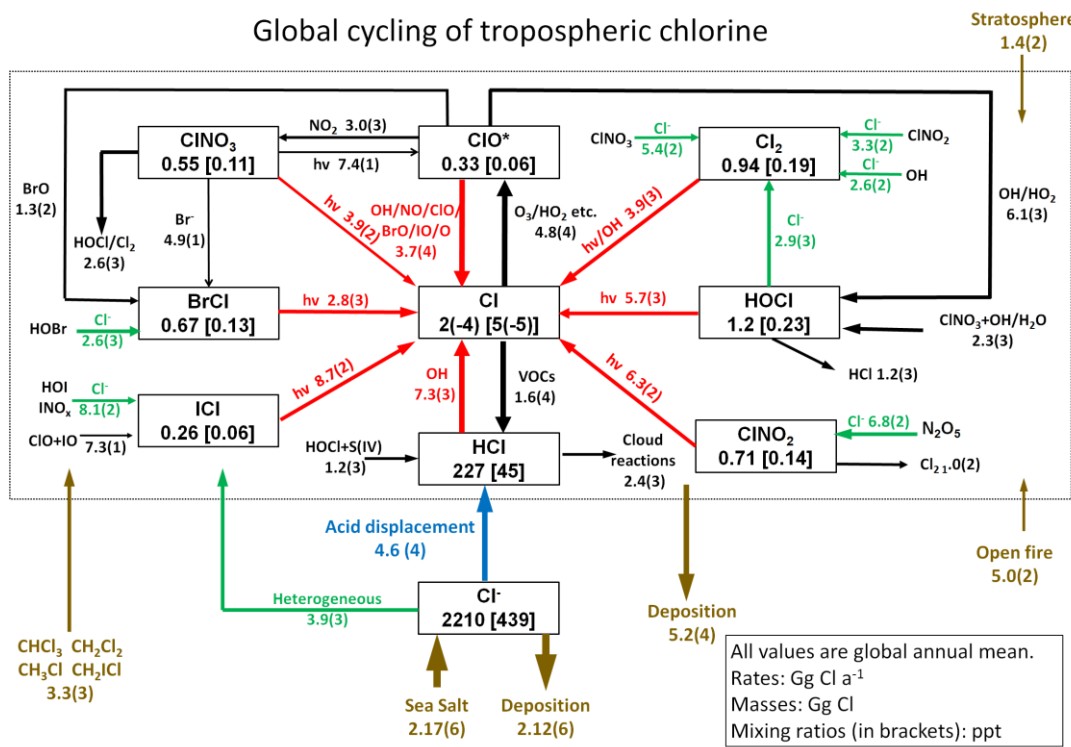

**Figure 1a. Global budget and cycling of tropospheric inorganic chlorine (Cl$_y$) in GEOS-Chem. Read 1.0(4) as 1.0x10$^4$. Reactions producing Cl atoms are in red. Heterogeneous reactions are in green. The dotted box indicates the Cl$_y$ family, and arrows into and out of that box represent general sources and sinks of Cl$_y$. Reactions with rate <100 Gg Cl a$^{-1}$ are not shown. ClO* stands for ClO + OClO + ClO$_2$ + 2Cl$_2$O$_2$; most is present as ClO. The superfast Cl-ClOO cycling is not included as it does not affect the Cl atom concentration.**

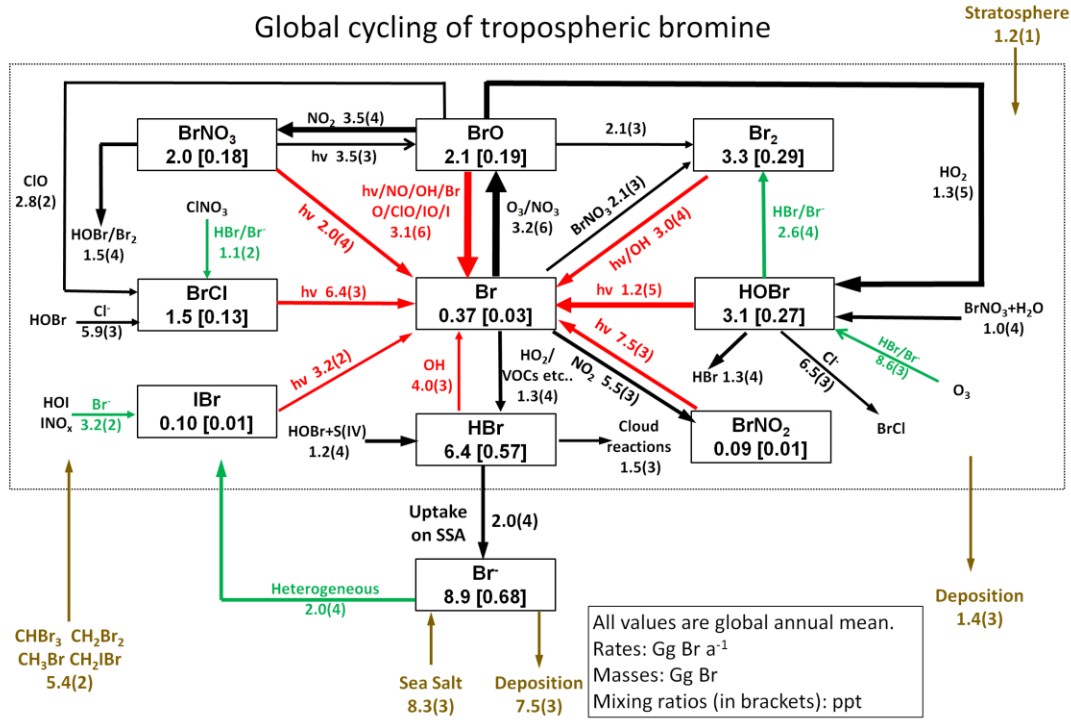

**Figure 1b. Same as Figure 1a but for tropospheric inorganic bromine (Br$_y$).**

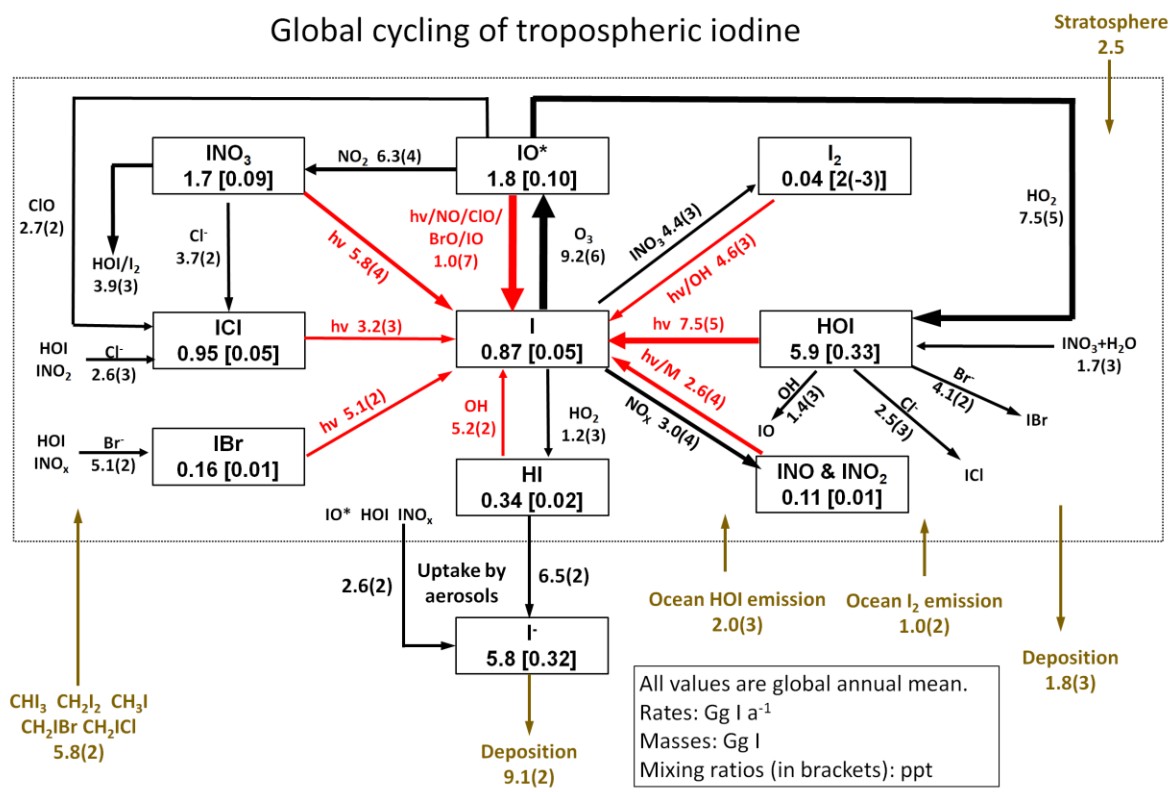

**Figure 1c. Same as Figure 1a but for tropospheric inorganic iodine ($I_y$). IO\* stands for IO + OIO + $2I_2O_2$ + $2I_2O_3$ + $2I_2O_4$.**

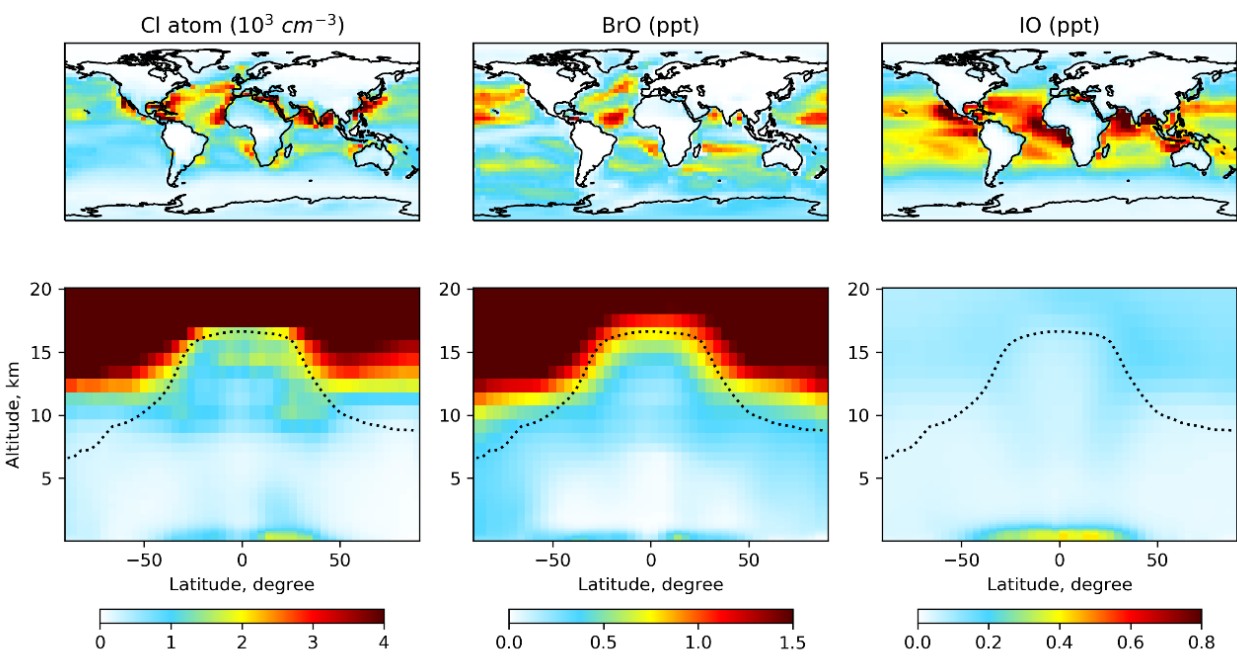


**Figure 2. Global distribution of annual mean GEOS-Chem number densities of Cl atoms and mixing ratios of BrO and IO. Upper panels show surface air values and lower panels show zonal means as a function of latitude and altitude. Dashed lines indicate the tropopause.**

## Global vertical distribution of reactive chlorine, bromine, and iodine gases

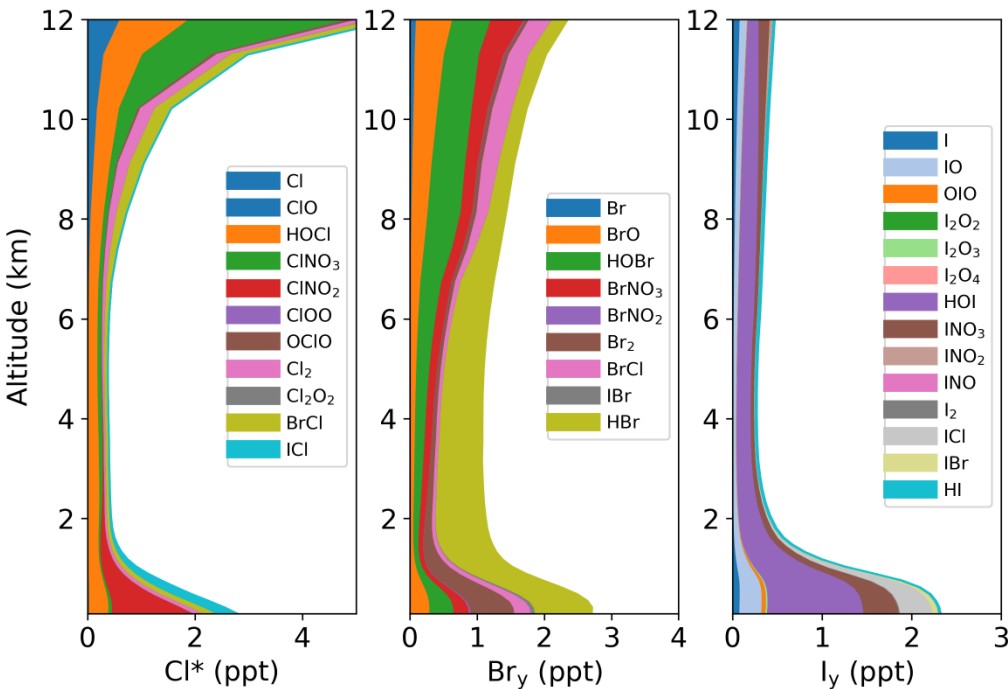

**Figure 3. Global annual mean vertical speciation in GEOS-Chem of reactive chlorine (Cl\* ≡ Cl$_y$ – HCl), gaseous inorganic bromine (Br$_y$),**
**and gaseous inorganic iodine (I$_y$, right).**

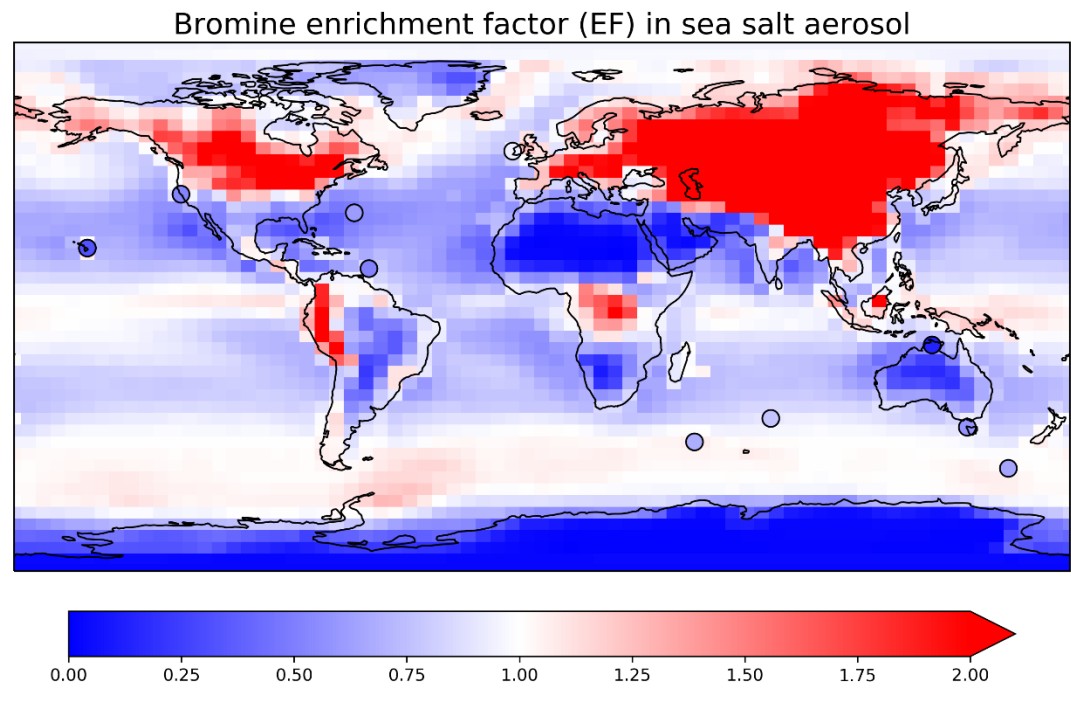

**Figure 4. Annual mean bromine enrichment factor (EF) of sea salt aerosol (SSA) in surface air. GEOS-Chem model results for total SSA**
**(contours) are compared to observations (circles). We sum [Br⁻] and [SSA] from both fine and coarse SSA and use equation (6) to**
**calculate EF.**

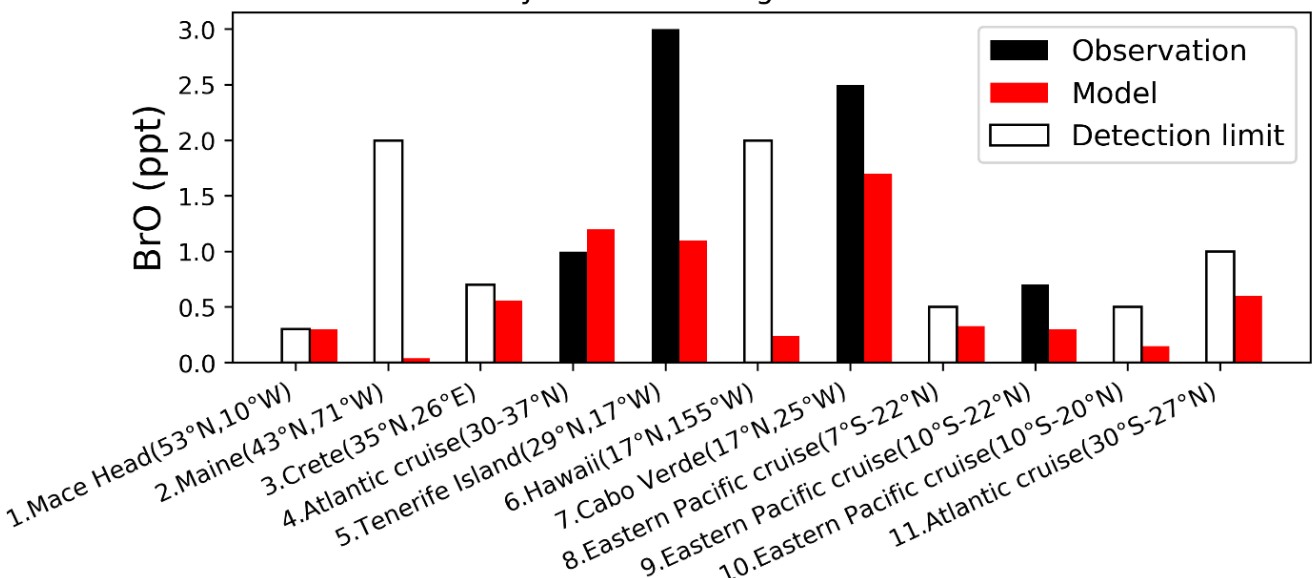

**Figure 5. Daytime surface air mixing ratios of BrO from island sites and ocean cruises, arranged from left to right in order of decreasing latitude. Observed values (black) are means for the reporting period in different years. Open bars show the measurement detection limit and indicate that the observation is below detection limit. Model values (red) are monthly mean values in 2016 taken for the same month and location as the observations. References: (1) Saiz-Lopez et al., 2004; Saiz-Lopez et al., 2006; (2) Keene et al., 2007; (3, 5, 6) Sander et al., 2003; (4, 11) Leser et al., 2003; Martin et al., 2009; (7) Read et al., 2008; Mahajan et al., 2010; (8, 9) Volkamer et al. (2010); (10) Volkamer et al. (2015).**

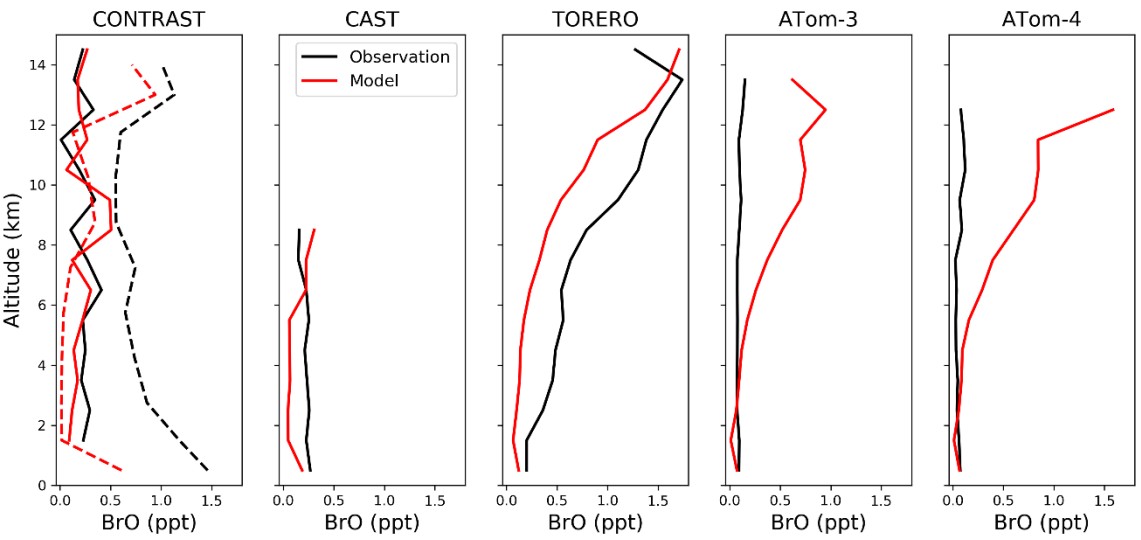

**Figure 6. Median vertical profiles of BrO mixing ratios from the CONTRAST (Jan-Feb 2014 over western tropical Pacific), CAST (Jan-Feb 2014 over western tropical Pacific), TORERO (Jan-Feb 2012 over eastern tropical Pacific), ATom-3 (Sep-Oct 2017 over Pacific and Atlantic), and ATom-4 (Apr-May 2018 over Pacific and Atlantic) campaigns. Observations are shown as medians in 1km vertical bins. Model values are shown as medians sampled along the flight tracks. There are two independent CONTRAST BrO data sets. The black solid line shows the CIMS data from Chen et al. (2016). The black dashed line shows the DOAS data from Koenig et al. (2017). The red solid and dashed lines show model values sampled along the flight tracks at the time of the available CIMS and DOAS observations respectively.**

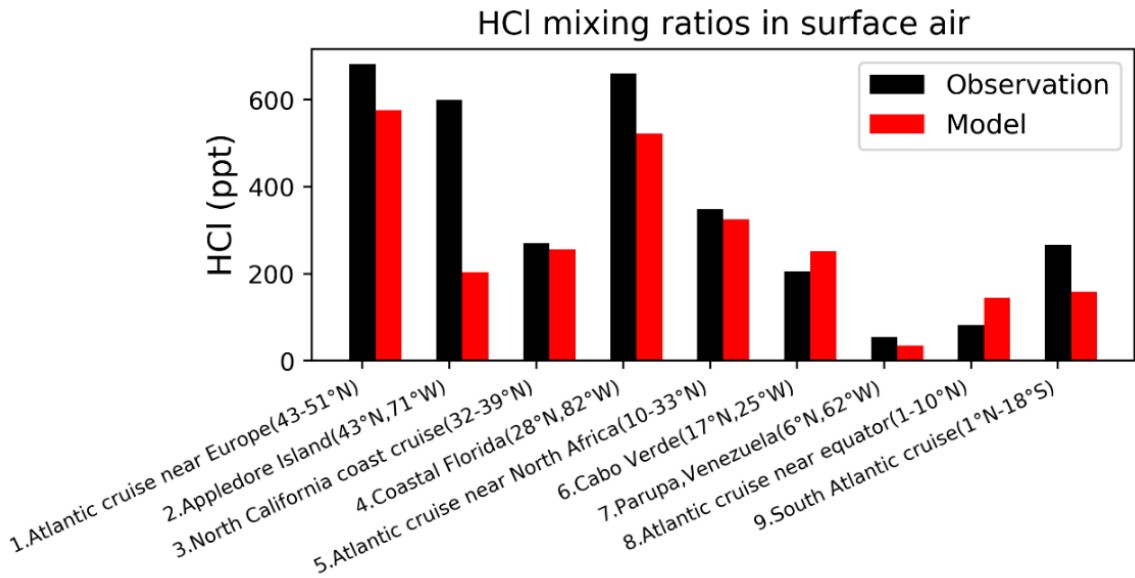


**Figure 7. Surface air mixing ratios of HCl at coastal and island sites and from ocean cruises, arranged from left to right in order of decreasing latitude. Observations (black) are means or medians depending on availability from the publications. Model values (red) are monthly mean values in 2016 taken for the same month and location as the observations. References: (1, 5, 8, 9) Keene et al. (2009); (2) Keene et al. (2007); (3) Crisp et al. (2014); (4) Dasgupta et al. (2007); (6) Sander et al. (2013); (7) Sanhueza and Garaboto (2002).**


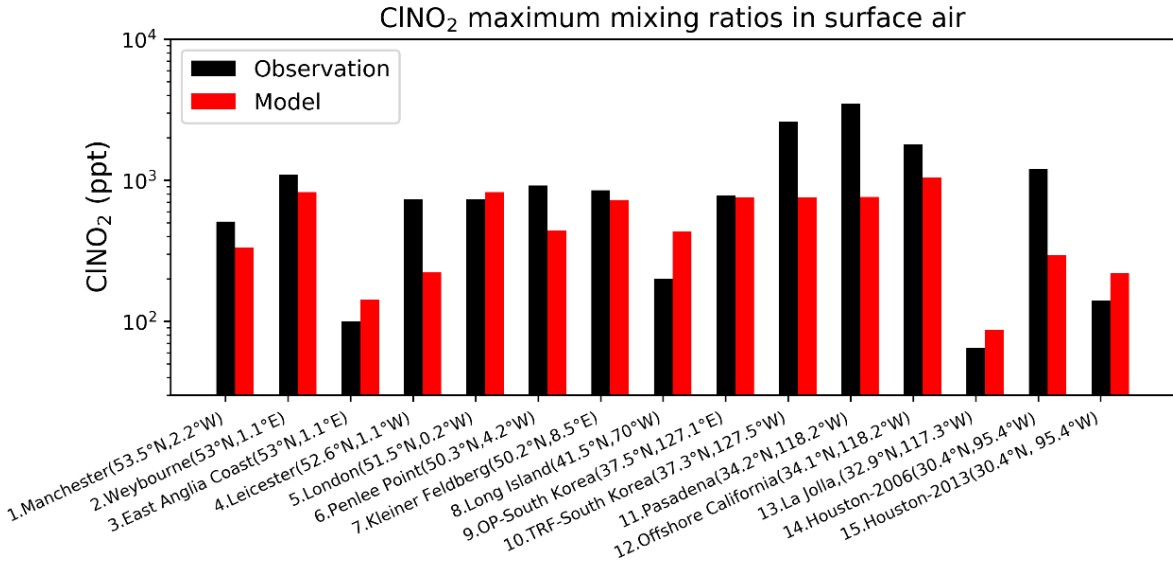

**Figure 8. Surface air mixing ratios of ClNO2 at coastal and island sites, arranged from left to right in order of decreasing latitude. Observed (black) and modeled (red) values are maxima for the reporting period. Model maxima are based on hourly values sampled at the same location and time period as the observations. References: (1) Priestley et al. (2018); (2,4,6) Sommariva et al. (2018); (3,5) Bannan et al. (2017); (7) Phillips et al. (2012); (58) Kercher et al. (2009); (9,10) Jeong et al. (2018); (11) Mielke et al. (2013); (12) Riedel et al. (2013); (13) Kim et al. (2014) (14) Osthoff et al. (2008) (15) Faxon et al. (2015).**


Median vertical profiles of chlorine species during WINTER

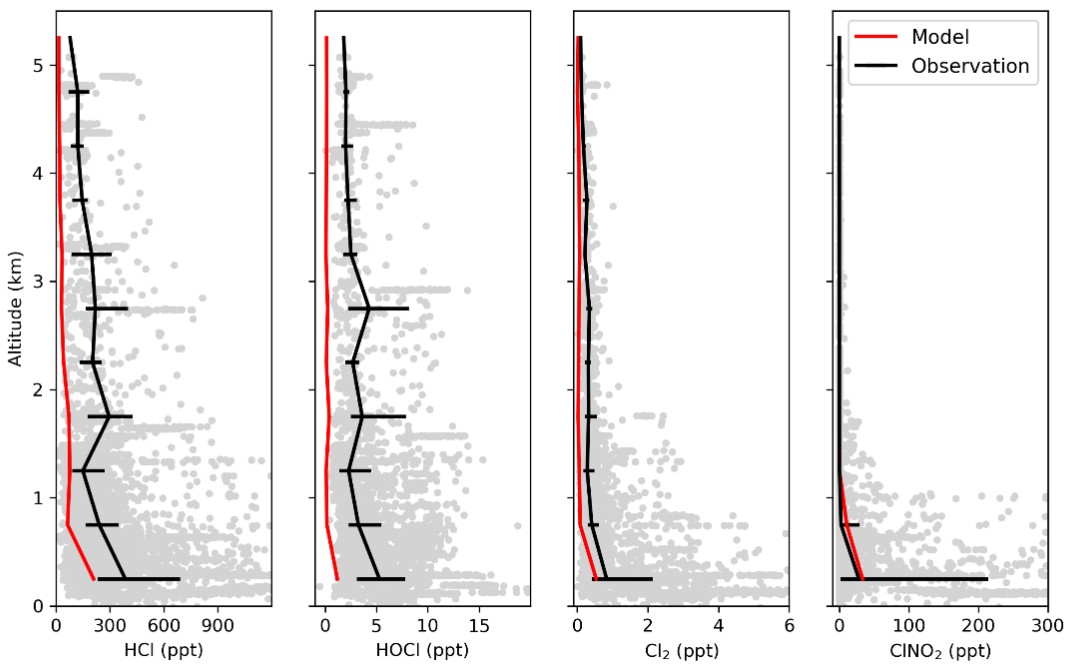

**Figure 9. Vertical profiles of HCl, HOCl, Cl₂, and nighttime ClNO₂ mixing ratios during the WINTER campaign over the eastern US and offshore in February–March 2015. Observations are shown as individual 1 min data points, with medians and 25th–75th percentiles in 500m vertical bins. ClNO₂ data exclude daytime (10:00–16:00 local) when mixing ratios are near zero in both the observations and the model. Model values are shown as medians sampled along the flight tracks.**

Median vertical profiles of chlorine species during ATom-3

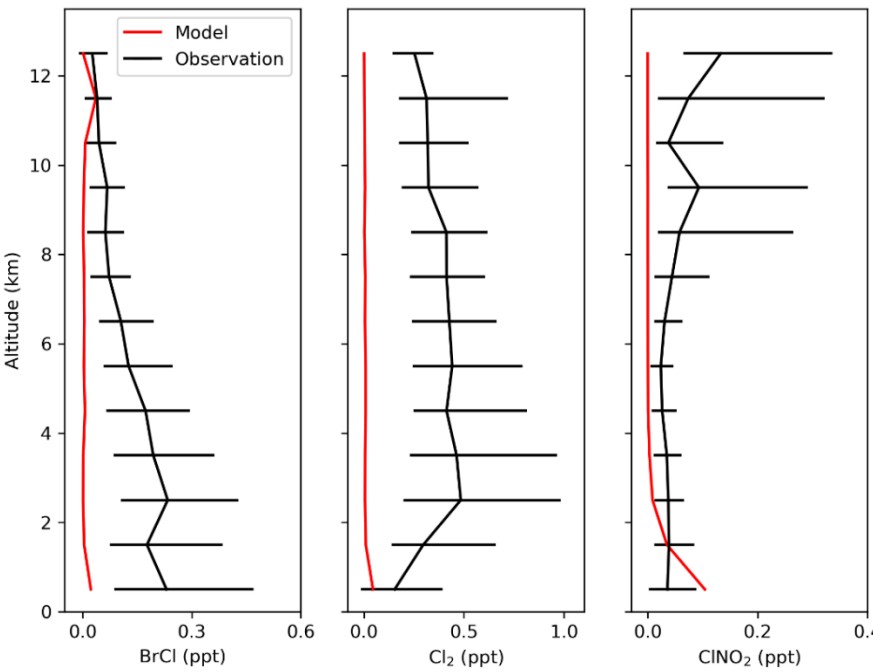

**Figure 10a. Vertical profiles of BrCl, Cl₂, and ClNO₂ mixing ratios from the ATom-3 campaign over the Pacific and Atlantic in September-October 2017. Observations are medians and 25th–75th percentiles in 1km vertical bins. Model values are medians sampled along the flight tracks.**

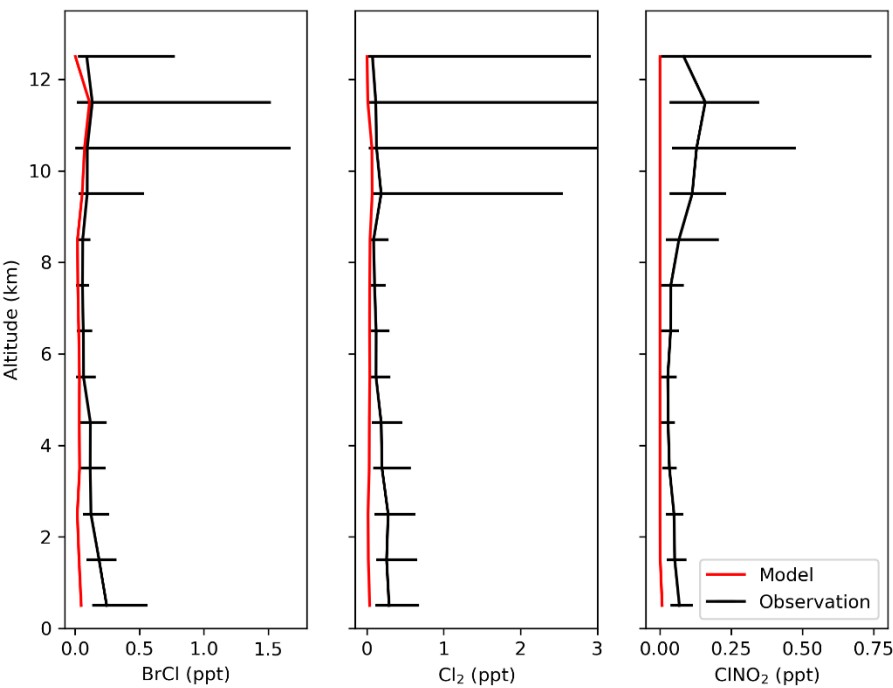

**Figure 10b. Same as Figure 10a but for the Atom-4 campaign in April-May 2018.**


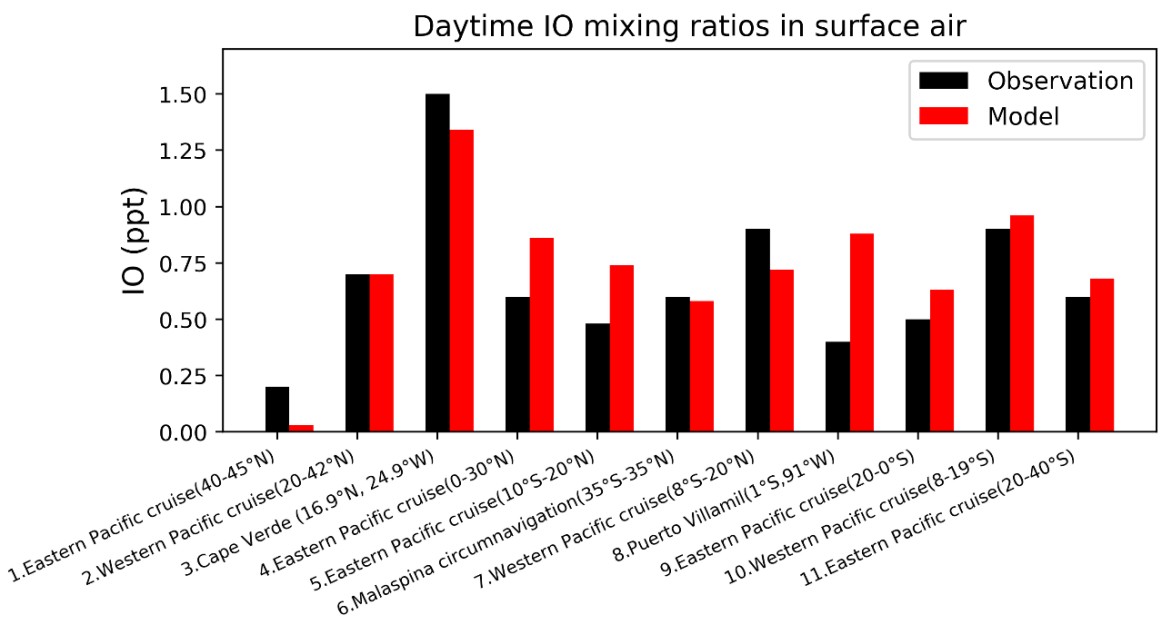

**Figure 11. Daytime surface air mixing ratios of IO from island sites and ocean cruises, arranged from left to right in order of decreasing**
**latitude. Observed values (black) are means for the reporting period in different years. Model values (red) are monthly mean values in 2016 taken for the same month and location as the observations. References: (1,4,9,12) Mahajan et al. (2012); (2,7,10) Großmann et al. (2013); (3) Mahajan et al. (2010); (5) Volkamer et al. (2015); (6) Prados-Roman et al. (2015); (8) Gómez Martín et al. (2013).**

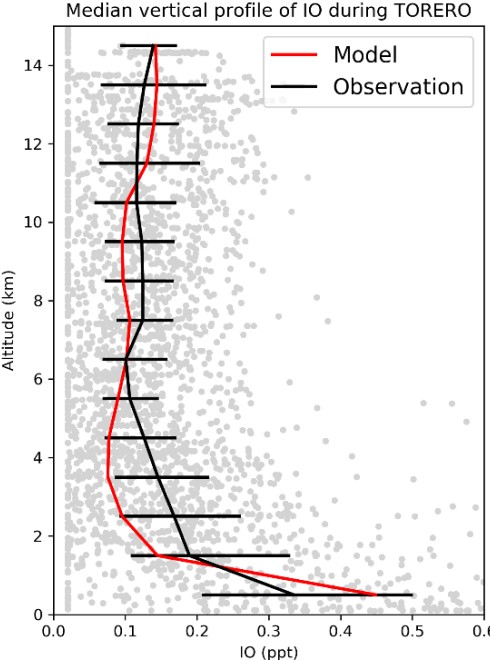

**Figure 12. Median vertical profile of IO mixing ratios from the TORERO (Jan-Feb 2012 over the eastern tropical Pacific) campaign. Observations are shown as individual 1 min data points, with medians and 25th–75th percentiles in 1km vertical bins. Model values are shown as medians sampled along the flight tracks.**

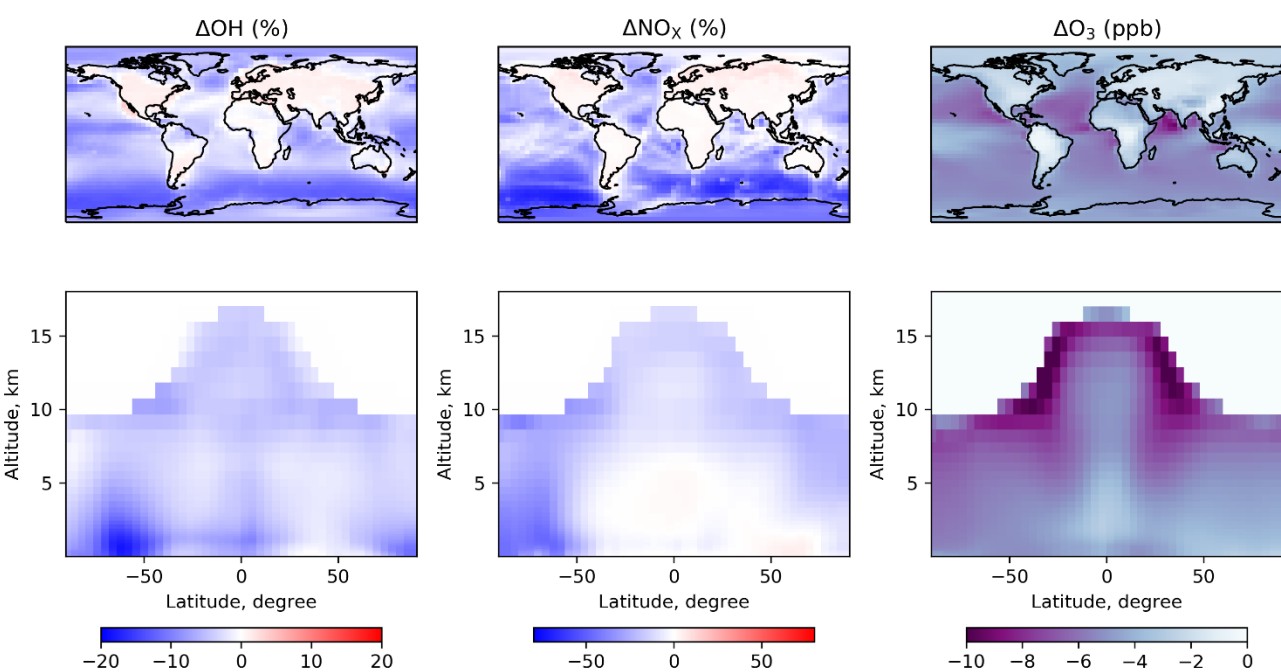


**Figure 13. Effects of halogen chemistry on tropospheric OH, NO$_x$, and ozone concentrations. The Figure shows differences in annual mean concentrations between the standard simulation and a sensitivity simulation removing all tropospheric halogen reactions. The top panels are for surface air and the bottom panels are for zonal means as a function of latitude and latitude. Only tropospheric grid boxes are shown.**

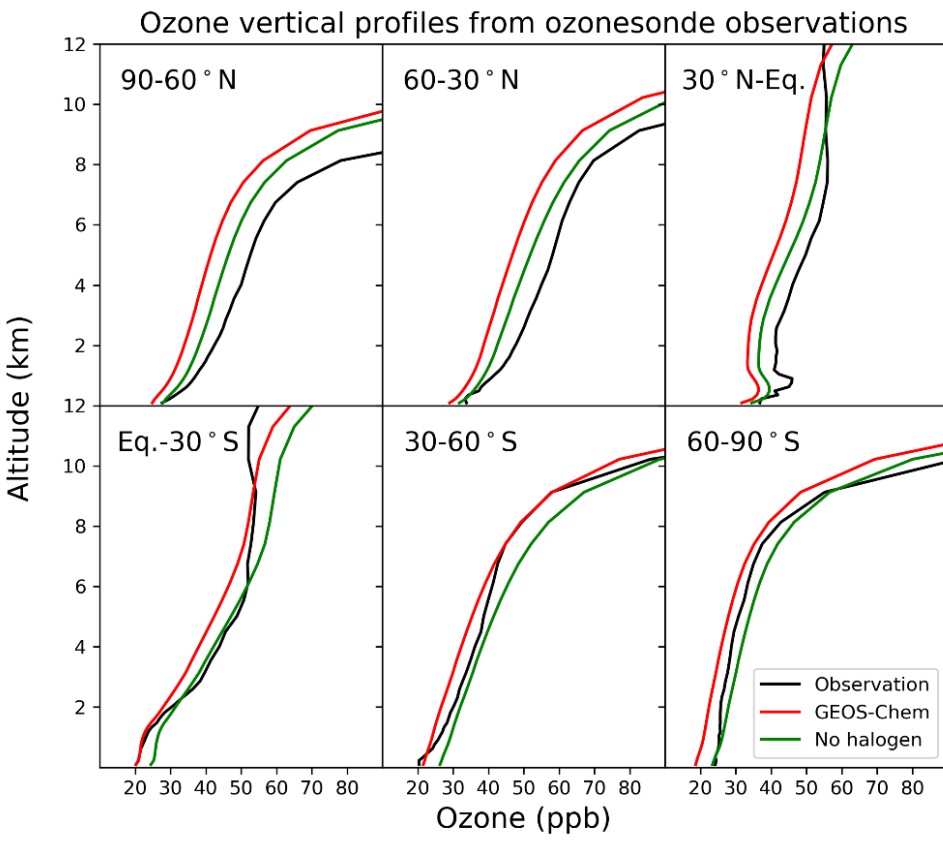

**Figure 14. Contributions of different pathways to the global annual loss of tropospheric ozone as a function of altitude. The "Other" pathway includes sinks from reactions with alkenes and HNO$_3$ deposition.**

**Figure 15. Annual average vertical profile of ozone mixing ratios from ozonesonde observations and the model for six zonal bands.**

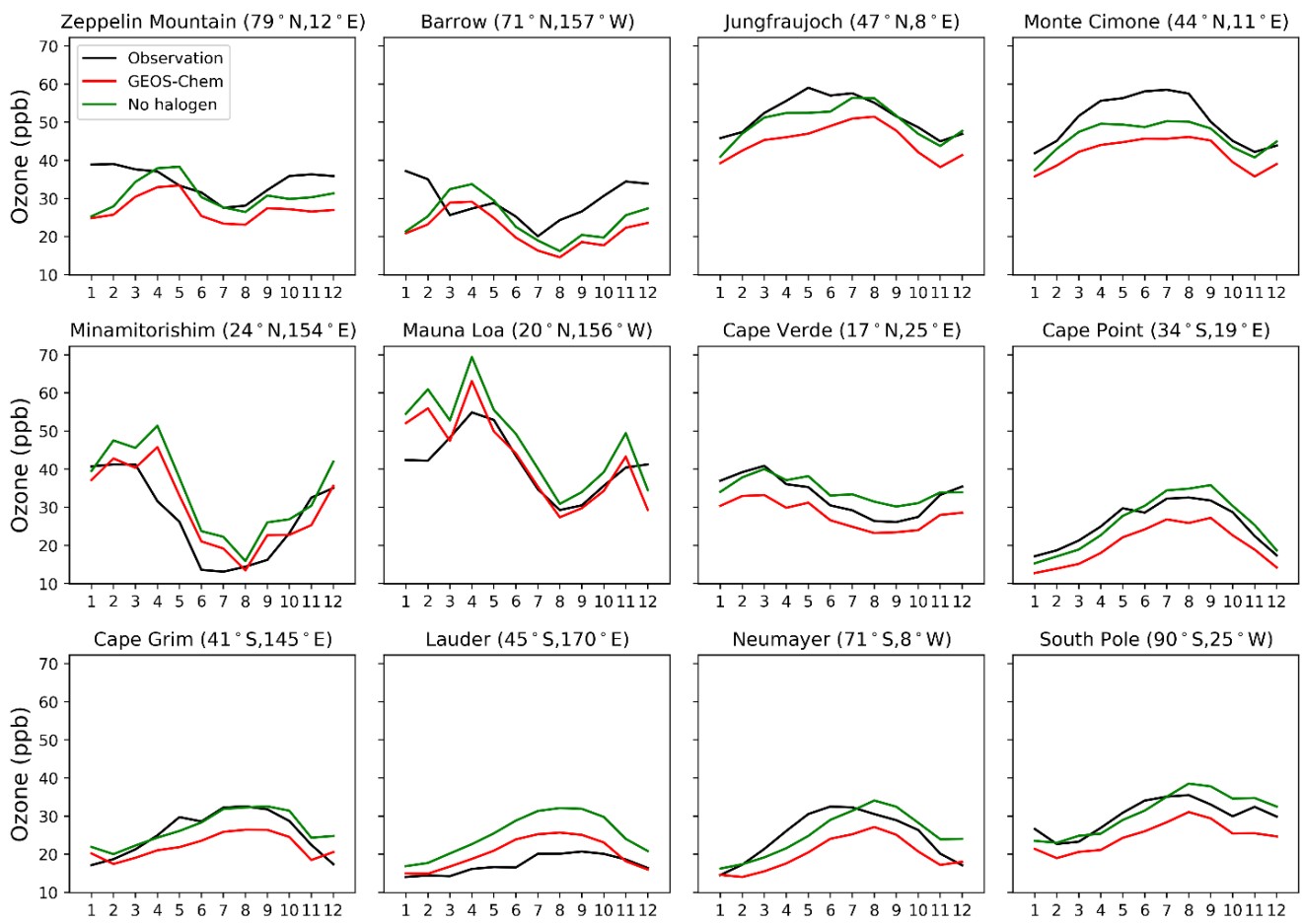

**Figure 16. Seasonal variation of surface ozone at a range of Global Atmospheric Watch (GAW) sites. Observational data are from the World Data Centre for Reactive Gases (WDCRG) and are 3-years monthly averages (2015–2017). Modeled values are monthly averages for 2016.**