# Peer review of "Global tropospheric halogen (Cl, Br, I) chemistry and its impact on oxidants"

_Atmospheric Chemistry and Physics, 2021_

## Referee Comment (RC1)

Wang et al. investigate the impact of halogens. The study is very interesting and I recommend publication in ACP after considering several minor changes as described below.

- p. 1, ll. 24-25: Change "less effective" to "less efficient".

- p. 2, l. 54: Change "examines" to "examine" (plural).

- p. 2, l. 57: There is no version 12.9 at `http://www.geos-chem.org`. Did you mean version 12.9.0?

- p. 2, l. 64: The terms $Cl_y$, $Br_y$, and $I_y$ have not been defined yet. Maybe refer to Tab. 1 for a definition?

- Bottom of p. 2: Why is $CHBr_3$ called "long-lived" and $CH_3Br$ "short-lived"? What are the lifetimes of these species in your model?

- p. 4: It is said that "Reactions (R3) and (R4) are important only in clouds because dissolution of $SO_2$ depends on the liquid water content." I think that the solution pH is another important factor. These reactions are not important in aerosols which have a low pH.

- p. 5, l. 152: I think that the texts S1 and S2 are important and concise enough to be included in the main text, instead of being hidden in the supplement.

- p. 6, l. 209: I find the statement "IO is higher in the upper troposphere" confusing because according to Figs. 2 and 3, most iodine is at the surface.

- p. 7, section 4: I think it would also be interesting to see the bromine enrichment factors (EF) in sea salt aerosols and a comparison to measurements. I am aware of the data presented by Zhu et al. (2019), however, after updating important reactions (as listed in Tab. 2), the results could be different now.

- p. 12: The Data availability section mentions that the model code is available at GEOS-Chem repository but only a general web page of the model is presented (`http://www.geos-chem.org`). Please provide the DOI where the exact version used in this study can be downloaded. Is `10.5281/zenodo.3950327` the correct DOI?

- Figs. 1a, 1b and 1c: Please explain what "etc." means next to the red arrow converting XO to X.

- According to the IUPAC Recommendations (page 1387 of Schwartz & Warneck "Units for use in atmospheric chemistry", Pure & Appl. Chem., 67(8/9), 1377-1406, 1995, `https://www.iupac.org/publications/pac/pdf/1995/pdf/6708x1377.pdf`) the usage of "ppb" and "ppt" is discouraged for several reasons. Instead, "nmol/mol" and "pmol/mol" should be used for gas-phase mole fractions. I suggest to replace the obsolete units.

---

## Referee Comment (RC2)

**Review of "Global tropospheric halogen (Cl, Br, I) chemistry and its impact on oxidants" by Wang *et al.**

**General comment:**

Wang *et al.* presented a description on tropospheric halogens (chlorine, bromine, and iodine) chemistry of an updated global chemical-transport model, GEOS-Chem, and assessed the effects of halogens on tropospheric oxidants and air pollutants. The paper has the potential to contribute to the increasingly recognized role of halogen chemistry in the troposphere. The topic of the manuscript also fits the scope of *Atmospheric Chemistry and Physics*. However, there are major concerns that should be addressed before it can be accepted for publication.

The biggest issue is the omission of anthropogenic (continental) source of reactive chlorine in the model, while there have been dozens of observations in the last decade suggested otherwise. Thornton et al. (2010) reported elevated levels of $ClNO_2$ at a continental site (~1400 km from the nearest coastline) in the U.S. Lee et al. (2018) observed high level of reactive chlorine species (HCl, $Cl_2$, $ClNO_2$, etc.) in the exhaust of coal-fired power plants in the U.S. Wang et al. (2016), Tham et al. (2016), Zhou et al. (2018), Yun et al. (2018), Peng et al. (2020), and many other recent studies in China consistently presented very high levels of $ClNO_2$ and other reactive chlorine species and almost all of these studies pointed to the anthropogenic source of chlorine. A recent report by Gunthe et al. (2021) suggests the existence of high loading of chloride in India.

The omission of anthropogenic chlorine resulted in many conclusions in the current manuscript that are not in line with previous observations, emission inventories, and model estimates which require further elaboration and/or adjustment.

(1) Line 66, a few emission inventories of anthropogenic have been proposed for China, including Liu et al. (2018), Fu et al. (2018), and Qiu et al. (2019). The anthropogenic chlorine in China alone could be up to ~0.5 Tg Cl $a^{-1}$, similar to the global biomass burning chlorine used here, so it's not "negligible". It's noteworthy that anthropogenic chlorine emission (mostly in the form of HCl, chloride) will be rapidly activated by anthropogenic $NO_x$ and form reactive chlorine, e.g., $ClNO_2$, while in the vast open ocean, the HCl from the acid displacement mostly reacts with OH to relase Cl atom with a slow rate.

(2) Line 174-175, "Cl atom concentrations are usually highest along polluted coastlines", while including anthropogenic chlorine source might lead to a different answer. In fact, a few modeling studies, including one by the same authors as the current paper, have shown that anthropogenic chlorine leads to much higher levels of chlorine species over continental area than those along the coast, e.g., Hossaini et al. (2016), Li et al. (2016), Li et al. (2020), Wang et al. (2020), etc.

(3) Line 245, it appears that the authors did not consider the continental observations of HCl (e.g, the ones in Fig 7 in Hossaini et al., 2016) when conducting the model

performance evaluation. Based on Fig. 2 of the paper, I would expect that the simulated HCl over land would be much lower than the corresponding observations.

(4) Line 250, the same for $ClNO_2$. the authors only picked the observations at island and coastal environments, while the vast available measurements in China were not mentioned. An earlier version of GEOS-Chem with very similar chlorine source and chemistry setup (Table 5 in Wang et al., 2019), however, only simulated ~10% of the observed level in southern China.

(5) Line 257, this paragraph is particularly puzzling to me. Lee et al. (2018) specifically pointed out that the reactive chlorine species observed during the WINTER campaign are tied to the power plant plumes. However, the authors claimed that "modeled HCl is lower than the observations but mostly within the calibration uncertainty", while the current GEOS-Chem model did not include any power plant source of chlorine. Does it mean that the natural sources of chlorine used here is too efficient?

(6) Line 289, the co-existence of anthropogenic chlorine and VOCs emission means that the role of chlorine atom in VOCs oxidation would be more important than what is reported here.

(7) Line 303, "surface NOx increases over the continents and this is due to $ClNO_2$ chemistry". If $ClNO_2$ chemistry is an important factor, a full representation of its formation process (including the source of chloride) is then desired.

**Specific comment:**

1. Line 70-74, define 'long-lived' and 'short-lived'. I wonder why '$CH_2Cl_2$' and '$CHCl_3$' are listed as long-lived species when they have a lifetime ~100 days. Is '$CH_2Cl$' a typo? Also, '$CHBr_3$' has a lifetime of ~20 days, '$CH_2Br_2$' ~130 days, and '$CH_3Br$' ~ 2 years.

2. Reaction (1) to (4) and Line 120. What are the numbers used here?

3. Line 128, What was the original value?

4. Line 154, what is the simulation period? 2015 to 2016 with the first year discarded as spin-up? Please specify.

5. Line 167, how was '6.3' calculated? Also, the sum of 6.3 Tg (heterogeneous source) and 46 Tg (acid displacement) is different from 50 Tg (in line 76).

6. Line 179, what are the numbers in Sherwen et al. (2016b) and Zhu et al. (2019)?

7. Line 188 "HOBr is now more likely to react with S(IV)" is not consistent with line 190 "59% of HOBr heterogeneous reactions are with $Br^-$ and $Cl^-$, and 41% are with S(IV)".

8. Line 195, It would be more informative if the distributions of SSA (both horizontal and vertical) are presented.

9. Please unify the ozone lifetime change in line 298 ('10%') and line 312 ('11%' )

**Technical comment:**

10. Line 79, remove the extra ')'.

11. Line 152, please unify the use of 'IONO' or 'INO$_2$' and 'IONO2' or 'INO$_3$'.

12. Line 165, 'Cl*' was defined in line 156.

13. Line 176, 'global zonal mean' should be 'global mean'?

14. Line 271, '4.2' should be '4.3'.

15. Fig 12, NO$_x$ is not an oxidant. Also, the color scale should be improved for OH and NO$_x$.

**Reference:**

Fu, X., Wang, T., Wang, S., Zhang, L., Cai, S., Xing, J. and Hao, J., 2018. Anthropogenic emissions of hydrogen chloride and fine particulate chloride in China. *Environmental science & technology*, *52*(3), pp.1644-1654.

Gunthe, S.S., Liu, P., Panda, U., Raj, S.S., Sharma, A., Darbyshire, E., Reyes-Villegas, E., Allan, J., Chen, Y., Wang, X. and Song, S., 2021. Enhanced aerosol particle growth sustained by high continental chlorine emission in India. *Nature Geoscience*, *14*(2), pp.77-84.

Hossaini, R., Chipperfield, M.P., Saiz- Lopez, A., Fernandez, R., Monks, S., Feng, W., Brauer, P. and Von Glasow, R., 2016. A global model of tropospheric chlorine chemistry: Organic versus inorganic sources and impact on methane oxidation. *Journal of Geophysical Research: Atmospheres*, *121*(23), pp.14-271.

Lee, B.H., Lopez- Hilfiker, F.D., Schroder, J.C., Campuzano- Jost, P., Jimenez, J.L., McDuffie, E.E., Fibiger, D.L., Veres, P.R., Brown, S.S., Campos, T.L. and Weinheimer, A.J., 2018. Airborne observations of reactive inorganic chlorine and bromine species in the exhaust of coal- fired power plants. *Journal of Geophysical Research: Atmospheres*, *123*(19), pp.11-225.

Li, Q., Zhang, L., Wang, T., Tham, Y.J., Ahmadov, R., Xue, L., Zhang, Q. and Zheng, J., 2016. Impacts of heterogeneous uptake of dinitrogen pentoxide and chlorine activation on ozone and reactive nitrogen partitioning: improvement and application of the WRF-Chem model in southern China. *Atmospheric Chemistry and Physics*, *16*(23), pp.14875-14890.

Li, Q., Badia, A., Wang, T., Sarwar, G., Fu, X., Zhang, L., Zhang, Q., Fung, J., Cuevas, C.A., Wang, S. and Zhou, B., 2020. Potential effect of halogens on atmospheric oxidation and air quality in china. *Journal of Geophysical Research: Atmospheres*, *125*(9), p.e2019JD032058.

Liu, Y., Fan, Q., Chen, X., Zhao, J., Ling, Z., Hong, Y., Li, W., Chen, X., Wang, M. and Wei, X., 2018. Modeling the impact of chlorine emissions from coal combustion and prescribed waste incineration on tropospheric ozone formation in China. *Atmospheric Chemistry and Physics*, *18*(4), pp.2709-2724.

Peng, X., Wang, W., Xia, M., Chen, H., Ravishankara, A.R., Li, Q., Saiz-Lopez, A., Liu, P., Zhang, F., Zhang, C. and Xue, L., 2020. An unexpected large continental source of reactive bromine and chlorine with significant impact on wintertime air quality. *National Science Review*.

Qiu, X., Ying, Q., Wang, S., Duan, L., Zhao, J., Xing, J., Ding, D., Sun, Y., Liu, B., Shi, A. and Yan, X., 2019. Modeling the impact of heterogeneous reactions of chlorine on summertime nitrate formation in Beijing, China. *Atmospheric Chemistry and Physics*, *19*(10), pp.6737-6747.

Sherwen, T., Schmidt, J.A., Evans, M.J., Carpenter, L.J., Großmann, K., Eastham, S.D., Jacob, D.J., Dix, B., Koenig, T.K., Sinreich, R. and Ortega, I., 2016. Global impacts of tropospheric halogens (Cl, Br, I) on oxidants and composition in GEOS-Chem. *Atmospheric Chemistry and Physics*, *16*(18), pp.12239-12271.

Tham, Y.J., Wang, Z., Li, Q., Yun, H., Wang, W., Wang, X., Xue, L., Lu, K., Ma, N., Bohn, B. and Li, X., 2016. Significant concentrations of nitryl chloride sustained in the morning: investigations of the causes and impacts on ozone production in a polluted region of northern China. *Atmospheric chemistry and physics*, *16*(23), pp.14959-14977.

Thornton, J.A., Kercher, J.P., Riedel, T.P., Wagner, N.L., Cozic, J., Holloway, J.S., Dubé, W.P., Wolfe, G.M., Quinn, P.K., Middlebrook, A.M. and Alexander, B., 2010. A large atomic chlorine source inferred from mid-continental reactive nitrogen chemistry. *Nature*, *464*(7286), pp.271-274.

Wang, T., Tham, Y.J., Xue, L., Li, Q., Zha, Q., Wang, Z., Poon, S.C., Dubé, W.P., Blake, D.R., Louie, P.K. and Luk, C.W., 2016. Observations of nitryl chloride and modeling its source and effect on ozone in the planetary boundary layer of southern China. *Journal of Geophysical Research: Atmospheres*, *121*(5), pp.2476-2489.

Wang, X., Jacob, D.J., Eastham, S.D., Sulprizio, M.P., Zhu, L., Chen, Q., Alexander, B., Sherwen, T., Evans, M.J., Lee, B.H. and Haskins, J.D., 2019. The role of chlorine in global tropospheric chemistry. *Atmospheric Chemistry and Physics*, *19*(6), pp.3981-4003.

Wang, X., Jacob, D.J., Fu, X., Wang, T., Breton, M.L., Hallquist, M., Liu, Z., McDuffie, E.E. and Liao, H., 2020. Effects of Anthropogenic Chlorine on PM2. 5 and Ozone Air Quality in China. *Environmental Science & Technology*, *54*(16), pp.9908-9916.

Yun, H., Wang, W., Wang, T., Xia, M., Yu, C., Wang, Z., Poon, S.C., Yue, D. and Zhou, Y., 2018. Nitrate formation from heterogeneous uptake of dinitrogen pentoxide during a severe winter haze in southern China. *Atmospheric Chemistry and Physics*, *18*(23), pp.17515-17527.

Zhou, W., Zhao, J., Ouyang, B., Mehra, A., Xu, W., Wang, Y., Bannan, T.J., Worrall, S.D., Priestley, M., Bacak, A. and Chen, Q., 2018. Production of N 2 O 5 and ClNO 2 in summer in urban Beijing, China. *Atmospheric Chemistry and Physics*, *18*(16), pp.11581-11597.

Zhu, L., Jacob, D.J., Eastham, S.D., Sulprizio, M.P., Wang, X., Sherwen, T., Evans, M.J., Chen, Q., Alexander, B., Koenig, T.K. and Volkamer, R., 2019. Effect of sea salt aerosol on tropospheric bromine chemistry. *Atmospheric Chemistry and Physics*, *19*(9), pp.6497-6507.

---

## Author Comment (AC1)

We thank the reviewers for their time and comments. We have made efforts to improve the manuscript accordingly, please find response for corresponding points below.

Reviewer #1 Rolf Sander

**Wang et al. investigate the impact of halogens. The study is very interesting and I recommend publication in ACP after considering several minor changes as described below.**

**p. 1, ll. 24-25: Change "less effective" to "less efficient".**

Changed.

**p. 2, l. 54: Change "examines" to "examine" (plural).**

Changed.

**p. 2, l. 57: There is no version 12.9 at http://www.geos-chem.org. Did you mean version 12.9.0?**

Yes, it means 12.9.0.

**p. 2, l. 64: The terms $Cl_y$, $Br_y$, and $I_y$ have not been defined yet. Maybe refer to Tab. 1 for a definition?**

We have followed the suggest and referred to Table 1 for the definition (line 65).

**Bottom of p. 2: Why is $CHBr_3$ called "long-lived" and $CH_3Br$ "short- lived"? What are the lifetimes of these species in your model?**

The lifetimes of $CHBr_3$ and $CH_3Br$ are 20 days and 1.5 years respectively in troposphere in the model. We have rephrased the sentences at line 69 and 71.

**p. 4: It is said that Reactions (R3) and (R4) are important only in clouds because dissolution of SO2 depends on the liquid water content." I think that the solution pH is another important factor. These reactions are not important in aerosols which have a low pH.**

We agree with the reviewer and added this point at line 134.

**p. 5, l. 152: I think that the texts S1 and S2 are important and concise enough to be included in the main text, instead of being hidden in the supplement.**

Texts S1 and S2 are now moved to the main text in Section 2.2 (line 159-166, 172-175).

**p. 6, l. 209: I find the statement "IO is higher in the upper troposphere" confusing because according to Figs. 2 and 3, most iodine is at the surface.**

This statement has been removed.

**p. 7, section 4: I think it would also be interesting to see the bromine enrichment factors (EF) in sea salt aerosols and a comparison to measurements. I am aware of the data presented by Zhu et al. (2019), however, after updating important reactions (as listed in Tab. 2), the results could be different now.**

We have added a new section 4.1 and Figure 4 to discuss the bromine enrichment factors.

**p. 12: The Data availability section mentions that the model code is available at GEOS-Chem repository but only a general web page of the model is presented (http://www.geos-chem.org). Please provide the DOI where the exact version used in this study can be downloaded. Is 10.5281/zenodo.3950327 the correct DOI?**

Yes, that is the correct DOI. The information has been added to the data availability section.

**Figs. 1a, 1b and 1c: Please explain what "etc." means next to the red arrow converting XO to X.**

Those figures have been updated to explicitly describe "etc.".

**According to the IUPAC Recommendations (page 1387 of Schwartz & Warneck "Units for use in atmospheric chemistry", Pure & Appl. Chem., 67(8/9), 1377-1406, 1995, https://www.iupac.org/publications/pac/pdf/1995/pdf/6708x1377.pdf) the usage of "ppb" and "ppt" is discour-aged for several reasons. Instead, "nmol/mol" and "pmol/mol" should be used for gas-phase mole fractions. I suggest to replace the obsolete units.**

Thanks for the suggestion. Since ppb and ppt are still standard conventional units used in the community, we feel that we can communicate more effectively with those units.

**Reviewer #2**

**General comment:**

**Wang *et al.* presented a description on tropospheric halogens (chlorine, bromine, and iodine) chemistry of an updated global chemical-transport model, GEOS-Chem, and assessed the effects of halogens on tropospheric oxidants and air pollutants. The paper has the potential to contribute to the increasingly recognized role of halogen chemistry in the troposphere. The topic of the manuscript also fits the scope of *Atmospheric Chemistry and Physics*. However, there are major concerns that should be addressed before it can be accepted for publication.**

**The biggest issue is the omission of anthropogenic (continental) source of reactive chlorine in the model, while there have been dozens of observations in the last decade suggested otherwise. Thornton et al. (2010) reported elevated levels of ClNO2 at a continental site (~1400 km from the nearest coastline) in the U.S. Lee et al. (2018) observed high level of reactive chlorine species (HCl, Cl2, ClNO2, etc.) in the exhaust of coal-fired power plants in the U.S. Wang et al. (2016), Tham et al. (2016), Zhou et al. (2018), Yun et al. (2018), Peng et al. (2020), and many other recent studies in China consistently presented very high levels of ClNO2 and other reactive chlorine species and almost all of these studies pointed to the anthropogenic source of chlorine. A recent report by Gunthe et al. (2021) suggests the existence of high loading of chloride in India.**

**The omission of anthropogenic chlorine resulted in many conclusions in the current manuscript that are not in line with previous observations, emission inventories, and model estimates which require further elaboration and/or adjustment.**

Thanks for the comments. We have added a paragraph in Section 2.1 (line 74-84) to discuss why we do not include anthropogenic source of reactive chlorine in this work. We also added several sentences at the beginning of Section 4.3 (line 281-284) to acknowledge that the model could underestimate in continental boundary layers due to no anthropogenic chlorine emissions. We have added the references that reviewer pointed in the comments to help these discussions.

**(1) Line 66, a few emission inventories of anthropogenic have been proposed for China, including Liu et al. (2018), Fu et al. (2018), and Qiu et al. (2019). The anthropogenic chlorine in China alone could be up to ~0.5 Tg Cl a-1, similar to the global biomass burning chlorine used here, so it's not "negligible". It's noteworthy that anthropogenic chlorine emission (mostly in the form of HCl, chloride) will be rapidly activated by anthropogenic NOx and form reactive chlorine, e.g., ClNO2, while in the vast open ocean, the HCl from the acid displacement mostly reacts with OH to relase Cl atom with a slow rate.**

We agree that anthropogenic chlorine emission in China is important regionally, but it is still small from a global budget perspective. Please check the above response and added text in Section 2.1 for details. Please also note that our study focuses on the global scale. A discussion of

anthropogenic chlorine in China with the same model framework has been published by us previously (Wang et al. 2020, cited in the text).

**(2) Line 174-175, "Cl atom concentrations are usually highest along polluted coastlines", while including anthropogenic chlorine source might lead to a different answer. In fact, a few modeling studies, including one by the same authors as the current paper, have shown that anthropogenic chlorine leads to much higher levels of chlorine species over continental area than those along the coast, e.g., Hossaini et al. (2016), Li et al. (2016), Li et al. (2020), Wang et al. (2020), etc.**

The context of this statement is the discussion of model result, so the sentence only describes the model results that anthropogenic inorganic chlorine sources are not included. We have added "simulated" to this sentence at line 197 to make it clearer.

**(3) Line 245, it appears that the authors did not consider the continental observations of HCl (e.g, the ones in Fig 7 in Hossaini et al., 2016) when conducting the model performance evaluation. Based on Fig. 2 of the paper, I would expect that the simulated HCl over land would be much lower than the corresponding observations.**

We agree with the reviewer. However, as discussed previously by Wang et al. (2019), using the only available global emission inventory of McCulloch et al. (1999) (the one used in Hossaini et al., 2016) will result in even larger biases. Therefore, we choose not to focus on continental emissions and observations. Please check the added text in Section 2.1 and 4.3 for details.

**(4) Line 250, the same for ClNO2. the authors only picked the observations at island and coastal environments, while the vast available measurements in China were not mentioned. An earlier version of GEOS-Chem with very similar chlorine source and chemistry setup (Table 5 in Wang et al., 2019), however, only simulated ~10% of the observed level in southern China.**

Please check the added text in Section 4.3. Please also note that this work is a global modeling study and we focus on the global scale. A discussion of chlorine in China with the same model framework has been published elsewhere (Wang et al. 2020 cited in the text).

**(5) Line 257, this paragraph is particularly puzzling to me. Lee et al. (2018) specifically pointed out that the reactive chlorine species observed during the WINTER campaign are tied to the power plant plumes. However, the authors claimed that "modeled HCl is lower than the observations but mostly within the calibration uncertainty", while the current GEOS-Chem model did not include any power plant source of chlorine. Does it mean that the natural sources of chlorine used here is too efficient?**

According to Lee et al. (2018), cited in the text, power plant related chorine species were only observed in several very short periods during WINTER. For the whole WINTER campaign, its contribution should be very small (< calibration uncertainty). Therefore, it is not likely that the current natural source of chlorine is too efficient based on this comparison.

**(6) Line 289, the co-existence of anthropogenic chlorine and VOCs emission means that the role of chlorine atom in VOCs oxidation would be more important than what is reported here.**

We agree with the reviewer and have added a sentence at line 328 in Section 5.1 to address this point.

**(7) Line 303, "surface NOx increases over the continents and this is due to ClNO2 chemistry". If ClNO2 chemistry is an important factor, a full representation of its formation process (including the source of chloride) is then desired.**

We have added sentences in line 342-346 to describe the process more clearly.

**Specific comment:**

**1. Line 70-74, define 'long-lived' and 'short-lived'. I wonder why 'CH2Cl2' and 'CHCl3' are listed as long-lived species when they have a lifetime ~100 days. Is 'CH2Cl' a typo? Also, 'CHBr3' has a lifetime of ~20 days, 'CH2Br2' ~130 days, and 'CH3Br' ~ 2 years.**

We have rephrased the sentences at line 69 and 71.

**2. Reaction (1) to (4) and Line 120. What are the numbers used here?**

The numbers are now added in the text (line 130-131).

**3. Line 128, What was the original value?**

The original value for $k_3^I$ is $3.2 \times 10^9$ $M^{-1}s^{-1}$, which is the upper limit in (Liu, 2000). We have included in line 140.

**4. Line 154, what is the simulation period? 2015 to 2016 with the first year discarded as spin-up? Please specify.**

Added in line 183.

**5. Line 167, how was '6.3' calculated? Also, the sum of 6.3 Tg (heterogeneous source) and 46 Tg (acid displacement) is different from 50 Tg (in line 76).**

This number 6.3 Tg Cl a$^{-1}$ is the sum of following processes: HOBr+Cl$^-$ (2.6 Tg Cl a$^{-1}$), HOCl+Cl$^-$ (1.5 Tg Cl a$^{-1}$), HOI/IONO$_x$+Cl$^-$ (0.8 Tg Cl a$^{-1}$), N$_2$O$_5$+Cl$^-$ (0.68 Tg Cl a$^{-1}$), ClNO$_3$+Cl$^-$ (0.27 Tg Cl a$^{-1}$), OH+Cl$^-$ (0.26 Tg Cl a$^{-1}$), ClNO$_2$+Cl$^-$ (0.16 Tg Cl a$^{-1}$).

This 6.3 Tg Cl a$^{-1}$ presents the Cl* generated from Cl$^-$ in clouds and aerosols, including those from dissolved HCl in liquid clouds. We have made this clearer in the text (line 189). In contrast, the 50 Tg only represents the mobilization of chlorine from sea salt aerosols.

**6. Line 179, what are the numbers in Sherwen et al. (2016b) and Zhu et al. (2019)?**

The tropospheric mean mixing ratio of BrCl is 0.69 ppt in Zhu et al. (2019). We have added this value in the text (line 203). There is no average number presented in Sherwen et al. (2016b), but the difference can be easily identified by comparing the Figure 5 in Sherwen et al. (2016b) and the Figure 3 in this manuscript.

**7. Line 188 "HOBr is now more likely to react with S(IV)" is not consistent with line 190 "59% of HOBr heterogeneous reactions are with Br$^-$ and Cl$^-$, and 41% are with S(IV)".**

We think the statement is consistent and the logic is correct here. In the text we first described the results in Zhu et al. (2019) and make the statement that HOBr is now more likely to react with S(IV) than in Zhu et al. (2019). We have made the sentence clearer in the text (line 211).

**8. Line 195, It would be more informative if the distributions of SSA (both horizontal and vertical) are presented.**

A figure (Figure S1) has been added to describe the distributions of SSA in the supplement.

**9. Please unify the ozone lifetime change in line 298 ('10%') and line 312 ('11%')**

Corrected.

**Technical comment:**

**10. Line 79, remove the extra ')'.**

Removed.

**11. Line 152, please unify the use of 'IONO' or 'INO2' and 'IONO2' or 'INO3'.**

Corrected. We now use $INO_2$ and $INO_3$ throughout the manuscript.

**12. Line 165, 'Cl*' was defined in line 156.**

The definition has been removed.

**13. Line 176, 'global zonal mean' should be 'global mean'?**

Changed.

**14. Line 271, '4.2' should be '4.3'.**

Corrected (typo).

**15. Fig 12, NOx is not an oxidant. Also, the color scale should be improved for OH and NOx.**

We have changed the title to "Halogen driven changes in OH, NOx, and ozone". We think the current color scale is fine as we want to show both positive and negative values clearly.

References:

Lee, B. H., Lopez-Hilfiker, F. D., Schroder, J. C., Campuzano-Jost, P., Jimenez, J. L., McDuffie, E. E., Fibiger, D. L., Veres, P. R., Brown, S. S., Campos, T. L., Weinheimer, A. J., Flocke, F. F., Norris, G., O'Mara, K., Green, J. R., Fiddler, M. N., Bililign, S., Shah, V., Jaeglé, L., and Thornton, J. A.: Airborne Observations of Reactive Inorganic Chlorine and Bromine Species in the Exhaust of Coal-Fired Power Plants, Journal of Geophysical Research: Atmospheres, 123, 11,225-211,237, 10.1029/2018jd029284, 2018.

Liu, Q.: Kinetics of aqueous phase reactions related to ozone depletion in the arctic troposphere: Bromine chloride hydrolysis, bromide ion with ozone, and sulfur(IV) with bromine and hypobromous acid, PhD thesis, Dep. of Chemistry, Purdue University, USA, 2000.

McCulloch, A., Aucott, M. L., Benkovitz, C. M., Graedel, T. E., Kleiman, G., Midgley, P. M., and Li, Y.-F.: Global emissions of hydrogen chloride and chloromethane from coal combustion, incineration and industrial activities: Reactive Chlorine Emissions Inventory, J. Geophys. Res.-Atmos., 104, 8391–8403, https://doi.org/10.1029/1999jd900025, 1999.

Sherwen, T., Schmidt, J. A., Evans, M. J., Carpenter, L. J., Großmann, K., Eastham, S. D., Jacob, D. J., Dix, B., Koenig, T. K., Sinreich, R., Ortega, I., Volkamer, R., Saiz-Lopez, A., Prados-Roman, C., Mahajan, A. S., and Ordóñez, C.: Global impacts of tropospheric halogens (Cl, Br, I) on oxidants and composition in GEOS-Chem, Atmospheric Chemistry and Physics, 16, 12239-12271, 10.5194/acp-16-12239-2016, 2016.

Wang, X., Jacob, D. J., Eastham, S. D., Sulprizio, M. P., Zhu, L., Chen, Q., Alexander, B., Sherwen, T., Evans, M. J., Lee, B. H., Haskins, J. D., Lopez-Hilfiker, F. D., Thornton, J. A., Huey, G. L., and Liao, H.: The role of chlorine in global tropospheric chemistry, Atmospheric Chemistry and Physics, 19, 3981-4003, 10.5194/acp-19-3981-2019, 2019.

Wang, X., Jacob, D. J., Fu, X., Wang, T., Breton, M. L., Hallquist, M., Liu, Z., McDuffie, E. E., and Liao, H.: Effects of Anthropogenic Chlorine on PM2.5 and Ozone Air Quality in China, Environ Sci Technol, 54, 9908-9916, 10.1021/acs.est.0c02296, 2020.

Zhu, L., Jacob, D. J., Eastham, S. D., Sulprizio, M. P., Wang, X., Sherwen, T., Evans, M. J., Chen, Q., Alexander, B., Koenig, T. K., Volkamer, R., Huey, L. G., Le Breton, M., Bannan, T. J., and Percival, C. J.: Effect of sea salt aerosol on tropospheric bromine chemistry, Atmos. Chem. Phys., 19, 6497-6507, 10.5194/acp-19-6497-2019, 2019.